# Molecular Characterization of Firework-Related Urban Aerosols using FT-ICR Mass Spectrometry

Qiaorong Xie[1,8], Sihui Su[2], Shuang Chen[2], Yisheng Xu[3], Dong Cao[4], Jing Chen[5], Lujie Ren[2], Siyao Yue[1,6,8], Wanyu Zhao[1,8], Yele Sun[1], Zifa Wang[1], Haijie Tong[6], Hang Su[6], Yafang Cheng[6], Kimitaka Kawamura[7], Guibin Jiang[4], Cong-Qiang Liu[2], and Pingqing Fu[2]

[1]State Key Laboratory of Atmospheric Boundary Layer Physics and Atmospheric Chemistry, Institute of Atmospheric Physics, Chinese Academy of Sciences, Beijing 100029, China
[2]Institute of Surface-Earth System Science, Tianjin University, Tianjin 300072, China
[3]State Key Laboratory of Environmental Criteria and Risk Assessment, Chinese Research Academy of Environmental Sciences, Beijing 100012, China
[4]State Key Laboratory of Environmental Chemistry and Ecotoxicology, Research Center for Eco-Environmental Science, Chinese Academy of Sciences, Beijing 100085, China
[5]School of Environmental Science and Engineering, Tianjin University, Tianjin, 300072, China
[6]Max Planck Institute for Chemistry, Multiphase Chemistry Department, Hahn-Meitner-Weg 1, 55128 Mainz, Germany
[7]Chubu Institute for Advanced Studies, Chubu University, Kasugai 487-8501, Japan
[8]College of Earth and Planetary Sciences, University of Chinese Academy of Sciences, Beijing 100049, China
*Correspondence to*: Pingqing Fu (fupingqing@tju.edu.cn)

**Abstract.** Firework (FW) emission has strong impacts on air quality and public health. However, little is known about the molecular composition of FW-related airborne particulate matter especially the organic fraction. Here we describe the detailed molecular composition of Beijing PM collected before, during, and after a FW event in New Year's Eve evening in 2012. Subgroups of CHO, CHON, and CHOS were characterized using ultrahigh resolution Fourier transform-ion cyclotron resonance (FT-ICR) mass spectrometry. These subgroups comprise substantial fraction of aromatic-like compounds with low O/C ratio and high degrees of unsaturation, some of which plausibly contributed to the formation of brown carbon in Beijing PM. Moreover, we found that the number concentration of sulfur-containing compounds especially the organosulfates increased dramatically during the FW event, whereas the number concentration of CHO and CHON doubled after the event, which were associated with multiple atmospheric aging processes including the multiphase redox chemistry driven by $NO_x$, $O_3$, and $^{\bullet}OH$. These findings highlight that FW emissions can lead to a sharp increase of high molecular weight compounds particularly aromatic-like substances in urban particulate matter, which may affect the light absorption properties and adverse health effects of atmospheric aerosols.

## 1 Introduction

The wide-spread haze pollution in China has aroused much attention due to its strong impacts on air quality, human health, and climate change (Ramanathan et al., 2001;Pöschl, 2005;Lelieveld et al., 2015;Thomason et al., 2018;Kaufman et al., 2002). The levels of haze pollution is strongly dependent on the source of haze particles, e.g. industry, coal combustion,

vehicle emissions, cooking and biomass burning (Sun et al., 2013;Zheng et al., 2005). Among different haze particle sources the FW emission can be expected to play an important role in urban air quality during festivals (Feng et al., 2012;Jing et al., 2014;Jiang et al., 2015;Tian et al., 2014). However, the chemical composition of FW-related aerosols especially the organic fraction is not well characterized.

There are a high number of pollutants released by FW burning, such as sulfur dioxide, nitrogen oxide, volatile organic compounds, and particles comprising inorganic materials (e.g. potassium and sulfate), and organic compounds (e.g. *n*-alkanes and PAHs) (Feng et al., 2012). They impose threats on human health (Sarkar et al., 2010) and can reduce visibility (Vecchi et al., 2008). Moreover, real-time chemical composition measurements illustrated that FW-related organics were mainly secondary organic material (Jiang et al., 2015). Nonetheless, all those studies primarily focused on the inorganic

chemical species and relatively low molecular weight (LMW) organic compounds, while little is known about the molecular-level characterization of high molecular weight (HMW) organic compounds in urban aerosols during FW events, which contains important chemical composition information of aerosols.

Water-soluble organic carbon (WSOC) is a ubiquitous component of atmospheric aerosols. A large proportion of water-soluble organic matter is composed of HMW organic compounds that contain a substantial fraction of heteroatoms (N, S, O)

(Lin et al., 2012a;Mazzoleni et al., 2012;Wozniak et al., 2008;Wang et al., 2016). Highly oxygenated molecules contain a wide range of chemical functional groups such as peroxides, hydroperoxides, carbonyls, and per- carboxylic acids (Lee et al., 2019). Organic acids in oxygen-containing species contribute significantly to aerosol acidity. Lots of nitro-aromatic compounds in relatively high molecular weight compounds, often observed in biomass burning aerosols, are potential contributors to light absorption (Laskin et al., 2015;Lin et al., 2015). Moreover, organosulfates substantially contribute to the

secondary organic aerosol (SOA) mass (Tolocka and Turpin, 2012), which plays an important role in exploring the formation pathway of SOA (Shang et al., 2016;Riva et al., 2015;Riva et al., 2016;Passananti et al., 2016). Meanwhile, because of their polar and hydrophilic nature, organosulfates can influence the hygroscopic properties of aerosols (Estillore et al., 2016). Hence, to characterize both the compound class and individual compound level of organic aerosols (OA) is important for exploring the formation mechanisms, physicochemical properties, and environmental effects of firework-

related aerosols. Moreover, the large amount of firework emission is an ideal event to understand the contribution of anthropogenic precursors to the formation of organic aerosols.

A molecular-level characterization of chemical constituents in firework-related aerosols is challenging because of their highly chemical complexity with numbers of compounds. Less than 10–20% of water-soluble organics, limited to LMW, can be characterized at a molecular level by a combination of gas chromatography-mass spectrometry (GC-MS) (Wang et al.,

2006), ion chromatography, and high performance liquid chromatography (HPLC) (Hong et al., 2004). Recently, Fourier transform ion cyclotron resonance mass spectrometry (FT-ICRMS), one of the ultrahigh-resolution mass spectrometers (UHRMS) with extremely high mass accuracy, has been successfully used to characterize complex organic mixtures of water-soluble organic matter in urban aerosols (e.g. Ohno et al., 2016;Qi and O'Connor, 2014;Lin et al., 2012a;Wozniak et al., 2008;Jiang et al., 2016;Mazzoleni et al., 2012;Kundu et al., 2012;Xie et al., 2020). However up to date, little is known

about the detailed molecular information in firework-related aerosols. FT-ICRMS can characterize compounds with molecular weight from 100 Da to 1000 Da, especially for HMW compounds. Moreover, compounds containing nitrogen, sulfur, and phosphorus atoms in the organic mixture can be identified by FT-ICRMS with high resolution (Hawkes et al., 2016).

In this study, the molecular-level composition of HMW organic compounds in urban aerosols collected in Beijing during the firework events of traditional Chinese New Year Eve and Spring Festival was investigated using a 15-tesla ultrahigh resolution FT-ICRMS. The chemical composition and number concentrations of CHO, CHON, and CHOS subgroups in FW- and non-FW-related aerosols were mainly discussed. In addition, the detailed molecular characteristics of CHNOS species and their volatility using a molecular corridor method will be present in another study.

## 2 Materials and methods

### 2.1 Aerosol sampling

Total suspended particle (TSP) sampling was conducted on the roof of a building (8 m above ground level) in the campus of the Institute of Atmospheric Physics, Chinese Academy of Sciences (39°58′28″ N, 116°22′13″ E), a representative urban site in the central north part of Beijing. TSP samples were collected on a 12-hour basis from 21$^{st}$ to 23$^{rd}$ of January 2012 (i.e.

sample ID: New Year's Eve daytime, NYE D, before the FW event; New Year's Eve nighttime, NYE N, during the FW event; lunar New Year's Day daytime, LNY D, after the FW event; lunar New Year's Day nighttime, LNY N), including episodes of short-term pollution raised by FW emissions. Detailed sample information is shown in Table 1. The 48-hour clustering air mass trajectories (Figure S1) show that all of them mainly originated from the northwest. All aerosol and field blank samples were collected using a high-volume air sampler (Kimoto, Japan) with pre-combusted (6 h in 450°C in a muffle furnace)

quartz filters (20 cm × 25 cm, Pallflex). After the sampling, the filters were stored in a refrigerator at −20°C until analysis.

### 2.2 Chemical component analysis

One punch (diameter: 24 mm) of each filter sample was sonicated in 10 mL ultrapure Millipore Q water for 20 min. The solution was then filtered with 0.22 μm hydrophilic PTFE filters (Anpel, China). The concentrations of water-soluble $SO_4^{2-}$, $NO_3^-$, and $Cl^-$, $NH_4^+$, $Na^+$, $K^+$, $Mg^{2+}$, and $Ca^{2+}$ were measured using an ion chromatography equipped with IonPac AS11HC

(Anion) and IonPac CS12 (Cation) chromatographic column systems (Dionex Aquion, Thermo Scientific, America). Concentrations of WSOC and total dissolved nitrogen (TDN) in the aerosol extracts were measured by TOC-L and TNM-L (Shimadzu, Japan). Water-soluble organic nitrogen (WSON) was calculated as the difference between TDN and the sum of water-soluble inorganic nitrogen (WSIN, including $NO_3^-$, $NO_2^-$ and $NH_4^+$) (Altieri et al., 2016). In addition, the loadings of OC/EC (Elemental carbon) and PAHs on filter samples were measured using a Sunset OC/EC analyzer (Sunset Laboratory

Inc., Model-4) and a gas chromatography-mass spectrometer (GC-MS), respectively. There were eighteen detected PAHs,

including phenanthrene (PHE), anthracene (AN), fluoranthene (FLU), pyrene (PYR), retene (RET), benz[a]anthracene (BaA), chrysene/triphenylene (CHR), benzo[b]fluoranthene (BbF), benzo[k]fluoranthene (BkF), benzo[e]pyrene (BeP), benzo[a]pyrene (BaP), perylene (PER), anthanthrene (ANT), indeno[cd]pyrene (IcdP), dibenzo[ah]anthracene (DahA), (a,l)dibenopyrene, 1,3,5-triphenylbenzene , benzo[ghi]perylene (BgP) and coronene (COR). More detailed information of the water-soluble ions, WSOC, OC/EC and PAHs analysis was given elsewhere (Yue et al., 2016;Ren et al., 2018;Fu et al., 2008).

### 2.3 FT-ICRMS measurement

Approximately 4.5 cm$^2$ of each filter was extracted three times with ultrapure Milli-Q water by sonicating for 10 min. The extract was combined and loaded onto a SPE cartridge (Oasis HLB, Waters, U.S.) for desalting, which had been preconditioned with methanol and Milli-Q water. The majority of inorganic ions, low molecular weight organic molecules, and sugars were not retained by the cartridge (Lin et al., 2012a). Then, the cartridge was washed with 5 mL Millipore Q water and dried under a nitrogen flow for one hour. Subsequently, the organic compounds retained on the cartridge were eluted using 12 mL of methanol to avoid incomplete elution. The eluate was immediately concentrated by a rotary evaporator and re-dissolved in 4 mL of methanol. The pretreated extracts were finally analyzed with a Bruker Solarix Fourier transform ion cyclotron resonance mass spectrometer (Bruker Daltonik, GmbH, Bremen, Germany) equipped with a 15.0 T superconducting magnet and an ESI ion source. Because the target species were water-soluble polar compounds, all the samples were analyzed in the negative ionization mode and infused into the ESI unit by syringe infusion at a flow rate of 120 μL h$^{-1}$. Ions were accumulated for 0.1 s in a hexapole collision cell. The mass limit was from 180 Da to 1000 Da. To enhance the signal-to-noise ratio and dynamic range, two hundred scans were averaged per spectrum. An average resolving power (m/Δm50%) of over 400 000 (at m/z = 400 Da) was achieved. Field blank filters were analyzed following the same procedure as the aerosol sample analysis. Other details of the experiment setup can be found elsewhere (Cao et al., 2016).

### 2.4 Molecular formula assignment

The mass spectra obtained by FT-ICRMS were internally recalibrated using an abundant homologous series of sulfur-containing organic compounds in the samples. Molecular formulae were assigned for peaks with a signal-to-noise (S/N) ratio >6 by allowing a mass error of 1.0 ppm between the measured and theoretically calculated mass. A Molecular Formula Calculator was used to calculate formulae in the mass range between 185 and 800 Da with elemental compositions up to 40 atoms of $^{12}$C, 100 of $^{1}$H, 40 of $^{16}$O, 2 of $^{14}$N, and 1 of $^{32}$S. The elemental ratio limits of H/C < 2.5, O/C < 1.2, N/C < 0.5, S/C< 0.2 and a nitrogen rule for even electron ions were used as further restrictions for formula calculation (Koch et al., 2007;Koch et al., 2005;Wozniak et al., 2008). Unambiguous molecular formula assignment was determined with the help of the homologous series approach for multiple formula assignments (Koch et al., 2007;Herzsprung et al., 2014). The isotopic peaks are removed in the present study. There were approximately 67−71% of the identified peaks were assigned in our samples. The intensities of them accounted for 70−75% of the total signal. The molecular formulas in blank filters with a

signal-to-noise ratio greater than that of the aerosol samples were subtracted from the real aerosol samples. In addition, because of the instrument limitations, the absolute mass concentration of each compound cannot be obtained. But, a semi-quantitative method can make up the defect to some extent by a normalized intensity, which has been applied in previous studies (Lin et al., 2012b;Jiang et al., 2016;Kourtchev et al., 2016;Dzepina et al., 2015).

## 2.5 Parameters calculation

To explore the saturation and oxidation degree of organic constituents of FW-related aerosols, we calculated the following useful parameters: double bound equivalent (DBE) and aromaticity equivalent ($X_c$), and carbon oxidation state ($OS_C$), respectively.

The value of DBE is calculated along the Eq. (1)

$$DBE = 1 + N_C - \frac{N_H}{2} + \frac{N_N}{2} \qquad (1)$$

where the $N_C$, $N_H$ and $N_N$ represent the number of C, H and N atoms in a molecular formula, respectively. Molecular formulae with DBE < 0 and formulae that disobey the nitrogen rule were discarded.

The value of $X_c$ is used to characterize aromatic and poly-aromatic compounds in highly complex compound mixtures. $X_c$ normally ranges from 0 to 3.0 and is calculated as follows (Yassine et al., 2014):

$$X_c = \frac{3(DBE - mN_O - nN_S) - 2}{DBE - mN_O - nN_S} \qquad (2)$$

If DBE $\leq$ $mN_O$ + $nN_S$, then $X_c$ = 0.

where m and n correspond to a fraction of O and S atoms involved in π-bond structures of a compound and are various for different functional groups. For instance, carboxylic acids, esters, and nitro functional groups have m = n = 0.5. When compounds containing functional groups such as aldehydes, ketones, nitroso, cyanate, alcohol, or ethers, m and n are adjusted to 1 or 0. Because ESI⁻ mode is the most sensitive to compounds containing carboxylic groups, we used m = n = 0.5 for the calculation of the $X_c$ in this study. $2.5 \leq X_c < 2.71$ indicates the presence of mono-aromatics and $X_c \geq 2.71$ indicates the presence of poly-aromatics.

The $OS_C$ is used to describe the composition of a complex mixture of organics undergoing oxidation processes. $OS_C$ is calculated for assignable molecular formulae using the Eq. (3) (Kroll et al., 2011):

$$OSc = -\sum_i OS_i \frac{n_i}{n_C} \qquad (3)$$

where $OS_i$ is the oxidation state associated with element $i$ and $n_i/n_C$ is the molar ratio of element $i$ to carbon within the molecule.

# 3 Results and discussion

## 3.1 Abundances of typical aerosol constituents

Table 1 presents that the mass concentrations of chemical components in the urban aerosol samples. The atmospheric abundances of inorganic ions such as $SO_4^{2-}$, $Cl^-$ and $K^+$ increased dramatically in the nighttime of the Chinese New Year Eve (the FW event), which were ten times higher than the non-FW periods, and then decreased sharply afterwards. $K^+$, an indicator generally used for biomass burning (Cheng et al., 2014), was the most abundant species among the measured ions and was about fifty times more than those during non-FW periods. Previous studies also reported that the concentrations of $K^+$ increased during FW events (Cheng et al., 2014;Tian et al., 2014). This is reasonable because that $K^+$ is a key component for the burst of FW. Similarly, $Cl^-$ also sharply increased during the FW event. But there was no influence of FW burning on $NO_3^-$. Some studies showed that the concentrations of $NO_3^-$ increased in the NYE nighttime (Zhang et al., 2017a), while others reported higher concentrations after the FW event (Yang et al., 2014b;Zhang et al., 2017a). These suggested that $KClO_4$ and $KClO_3$ are the main components of the FW emission, though $KNO_3$ is also the principal oxidizer in black powder (Wang et al., 2007). Both $SO_4^{2-}$ and $NO_3^-$ are secondary inorganic ions; such diversity may be due to the changes in emission sources. An increase of $SO_4^{2-}$ is associated with orange flames from lots of FW burning (Moreno et al., 2007). Fossil fuel combustion and vehicle emissions have been reported as important sources of $NO_3^-$ in Beijing (Ianniello et al., 2010;Wang et al., 2014), while these sources minimized due to a sharp decline in the population and vehicle; most of the people leave Beijing for their hometowns during the Spring Festival (Yang et al., 2014a;Zhang et al., 2017b). In addition, the concentrations of $Mg^{2+}$ and $Ca^{2+}$ were slightly higher in the NYE nighttime than the non-FW periods. They were mainly in the coarse particle mode (Huang et al., 2013;Xu et al., 2015).

Moreover, the mass concentrations of OC and EC during the FW event (sample NYE N) doubled those during non-FW periods, particularly for EC. Simultaneously, the WSOC concentration peaked sharply in the NYE nighttime. Moreover, the WSOC/OC ratio was higher during the FW period than non-FW periods, indicating more water-soluble OC was formed during the FW event. Compared to the non-FW period, the total concentration of eighteen detectable PAHs ($\Sigma18$ PAHs, PAH types were listed in Materials and methods) significantly increased for four times during the FW event, agreeing with the urban aerosol study by Kong et al. (2015), which found that FW burning was an important source for PAHs in Nanjing $PM_{2.5}$ during Spring Festival period in 2014. Furthermore, the detail molecular composition of WSOC components were characterized by ESI FT-ICRMS and discussed below.

## 3.2 General molecular characterization of organic aerosols

The reconstructed mass spectra of all samples by ESI FT-ICRMS are exhibited in Figure 1. The peak intensity is mainly affected by the initial concentration and ionization efficiency of the neutral compound (Lin et al., 2012a). ESI is sensitive to polar compounds, and the compounds reported in this study is easily ionized in the negative ion mode (Qi et al., 2020). On this basis and considering the fact that the spectra of all samples were obtained under the same ESI-MS condition, the peak

intensities of the same ions could be compared among different samples by assuming that matrix effects were relatively constant (Kourtchev et al., 2016;Lin et al., 2012a). To make a comparison among different spectra, the most arbitrary abundant $C_{18}H_{29}O_3S_1^-$ (m/z 325.18429) ion in NYE N sample ($2.2 \times 10^8$ arbitrary units), was defined as 100% (1 unit in reconstructed mass spectra in Figure 1); all peak intensities in the measured samples were normalized to it.

Thousands of formulae (~6000–9500) were obtained in each spectrum with the majority ranged from 150 to 700 Da. The molecular weights of formulae with high intensity primarily distributed between 300 and 400 Da, which were higher than previous studies with 200 - 300 Da. On the one hand, the compounds being explored in present study have a larger mass range; on the other hand, it was worth noting that some fractions of compounds might be lost during our sampling preparation, particularly for the low molecular weight ones. The formulae of different molecules are classified into CHO,

CHNO, CHNOS, and CHOS compounds. For example, CHNOS compounds refer to formulae that contain carbon, hydrogen, oxygen, nitrogen, and sulfur elements.

    The relative number abundances of compounds in four classes are shown in Figure 1. The CHO and CHNO compounds accounted for 50−71% in all four categories, while sulfur-containing compounds account for less. The total intensities of CHO and CHNO compounds were also dominant, accounting for 43−72%, which demonstrate that these categories of

compounds are abundant in both the number and mass concentrations in the urban aerosols. Nonetheless, the average number and intensity contributions of sulfur-containing compounds were 32% and 33% during the non-FW periods; they increased to 51% and 57% in the NYE nighttime, respectively, suggesting that FW emissions contribute significantly to sulfur-containing compounds.

    Tables 2, S1 and S2 show the number of compounds in each class and arithmetic and weighted mean elemental ratio for

them in each sample. When affected by the FW emissions, the number of compounds increased to 6836 in the NYE nighttime and 9511 in the LNY daytime in comparison with 5854 in the NYE daytime. Moreover, their average molecular weight increased from 405 ± 89 Da in in the NYE daytime to 439 ± 99 Da in the NYE nighttime and 448 ± 97 Da in the LNY daytime. These results suggested that FW emission contributes the formation of relatively high molecular weight compounds in urban aerosols. In addition, the average DBE values, an indicative of degree of unsaturation, increased from

9.35 ± 4.01 in the NYE daytime to 10.1 ± 4.82 in the NYE nighttime and 11.2 ± 4.98 in the LNY daytime. Compounds with low O/C and H/C ratios and high DBE values are likely to be aromatic-like species (Tong et al., 2016;Kourtchev et al., 2016), indicating that FW burning plays a significant role to the formation of these compounds. There was a similar trend for the intensity weighted mean elemental ratios of compounds with lower O/$C_w$ and H/$C_w$, and higher $DBE_w$. It is worth noting that the detected compounds showed lower O/C and H/C ratios and higher DBE values than those in previous studies (Table S3),

suggesting more aromatic compounds in the FW influenced aerosols in urban Beijing. More importantly, FW emission dramatically increased the amounts of HMW (>400 Da) organic compounds from 3022 compounds in the NYE daytime to 4264 compounds in the NYE nighttime and 5206 compounds in the LNY daytime, while the relative abundance of three categories compounds were different.

### 3.3 CHO compounds

CHO compounds detected in the ESI negative mode potentially include carboxyl and/or hydroxyl functional groups deprotonation effect (Cech and Enke, 2001). As shown in Table 2, there was no significant change for the number of CHO species between NYE N (2260 compounds) and NYE D (2045 compounds). Except for common compounds in two samples, the number and total intensities of the unique compounds in NYE N sample (591 compounds) were slightly increased compared with those only in NYE D sample (376 compounds) (Figure 2). However, the unique compounds increased considerably after the FW event, that is, in the LNY daytime (Table S1) with their number being up to 3120. On the one hand, the precursors emitted by FW burning at NYE night possibly produced a large number of CHO compounds in the LNY daytime under the photochemical reaction; on the other hand, they were affected by the spread of the regional FW emission of pollutants in the surrounding regions of Beijing during the Chinese New Year Eve. In addition, the production efficiency of oxygen-containing compounds during the day through photochemistry should be more significant than that at night.

As shown in Figure 3 and Figure S2, CHO compounds had $O_1$ to $O_{15}$ subgroups, which were classified by the number of O atoms in their molecules. As for $O_1$–$O_8$ subgroups, both the number and the intensity of them increased as the oxygen content increases, while they decreased from $O_9$ to $O_{15}$ subgroups. Among them, $O_4$–$O_{10}$ subgroups dominated the total of CHO compounds, and the number and the intensity of them accounted for 65–79% and 64–85% of the total compounds, respectively. After the FW event, the abundance of each $O_n$ subgroup considerably increased in the LNY daytime, particularly for the $O_{>7}$ subgroups, highlighting the importance of photochemical formation.

As shown in Figure 4, the high intensity CHO compounds in the fraction of water-soluble organic matter in urban aerosols are primarily with C numbers of 15–27 and DBE values of 6–15, indicating that they potentially have one or more benzene rings in their molecules. The DBE values and C numbers of CHO compounds in NYE N and LNY D samples vary in the ranges of 0–29 and 6–40, respectively, higher than those of 0–22 (DBE) and 7–35 (C number) for other samples. There were many HMW CHO compounds with a high degree of unsaturation in the FW effected aerosols. Moreover, they are high oxygen-containing compounds with O atoms more than 8 O atoms, which are potentially the highly oxidized and condensed aromatic compounds. In addition, some high intensity of compounds with low DBE values and O atoms were high in the FW effected aerosols, such as $C_{16}H_{32}O_2$, $C_{18}H_{36}O_2$, $C_{20}H_{40}O_2$, $C_{22}H_{44}O_2$, and $C_{24}H_{48}O_2$. They have an even carbon advantage, which could be fatty acids (Li et al., 2018;Kang et al., 2017;Fan et al., 2020).

CHO compounds with aromatic index (AI) > 0.5, a characteristic of condensed aromatic ring structures, were also more abundant in the LNY daytime (777 compounds) than in the NYE nighttime (484 compounds) (Table 3). The H/C and O/C ratios of CHO compounds in different samples with various AI regions were shown in Figure 5. Obviously, compounds with AI > 0.5 had low H /C ratios (< 1). The majority of them have DBE values above 7, indicating that they correspond to oxidized aromatic compounds, which are primarily of anthropogenic origin (Tong et al., 2016). Moreover, more species of them fall into the area of AI > 0.5 in the LNY daytime. This suggests that the pollutants emitted by FW burning may be

oxidized into aromatic CHO compounds under the oxidation by nighttime chemistry, while the photochemical reaction during the day is more efficient.

Different families of compounds with heteroatoms (e.g. O, N, S) overlap in terms of DBE, which may be inadequate to explain the level of unsaturation of organic compounds and to identify whether a molecular formula potentially has a (poly-) aromatic structure or not (Kourtchev et al., 2016;Yassine et al., 2014;Tong et al., 2016;Reemtsma, 2009). For instance, divalent atoms such as oxygen and sulfur do not influence the value of DBE, but they may contribute to the potential double bonds of that molecule. Unlike the parameter AI, the use of parameter $X_c$ can avoid this problem and help to more precisely identify and characterize aromatic and condensed aromatic compounds in highly complex WSOC mixtures (Yassine et al., 2014). The H/C and O/C ratios versus the MW and $X_c$ under different samples are shown in Figure S3. There are much more formulae in samples with an Xc > 2.5 (indicative of aromatics compounds) when using the $X_c$ classification than AI due to a large fraction of alkylated aromatics in the present study, which would be wrongly assigned as non-aromatics by AI (Kourtchev et al., 2016). The highest number of the aromatic compounds in the samples were observed for formulae with a pyrene core structure ($X_c$ = 2.83). The number of compounds with an ovalene core structure ($X_c$ = 2.92) and highly condensed aromatic structures or highly unsaturated ($X_c$ > 2.93) significantly increased by the FW burning event till the LNY daytime, suggesting the importance of photochemical oxidation (Figure S3).

$OS_C$ is an ideal parameter to describe the oxidation processes of a complex mixture of organics. Figure 6 shows overlaid OSc symbols for CHO compounds in NYE D, NYE N, and LNY D samples. Because of the direct and indirect influence by FW emissions, $OS_C$ shifted towards a less oxidized state with carbon atoms more than fifteen carbon atoms in NYE N and LNY D samples. The difference in $OS_C$ becomes even more significant with the increased number of C in the detected CHO compounds. As shown in Figure 6, different $OS_C$ value and C number indicate different types of compounds as previously characterized by Kroll et al. (2011). The semi-volatile and low-volatility oxidized organic aerosol (SV-OOA and LV-OOA) have the values of $OS_C$ between −1 and +1 and carbon atoms less than 13, which are associated with that are produced by multistep oxidation reactions. The biomass burning organic aerosol (BBOA) has lower $OS_C$, with $OS_C$ between −0.5 and −1.5 and carbon atoms more than 7. The molecules with $OS_C$ less than −1 and carbon atoms more than 20 might be associated with hydrocarbon-like organic aerosol (HOA).

More compounds with long carbon chains are found in aerosols affected by the FW emission. A large number of compounds with high peak intensities have similar $OS_C$ to the SV-OOA, while they have longer carbon chains, from $C_{15}$ to $C_{30}$ in the NYE N sample. Another important part of the FW-affected ions in NYE N and LNY D samples fall into the category of the BBOA, which are associated with primary particulate matter directly emitted into the atmosphere. Moreover, unlike compounds before the FW event, there were plenty of molecular formulae with low $OS_C$ in the area of HOA in both NYE N and the LNY D samples, which were possibly aromatic-like compounds.

## 3.4 CHON compounds

The trend of CHON species was similar to that of CHO (Figure 2). Although massive FW emissions occurred in the NYE nighttime, the number and total intensities of CHON and CHO showed minor changes between daytime and nighttime, while both of them clearly increased in the following LNY daytime. Their average molecular weights were $445 \pm 100$ Da in the
NYE nighttime and $472 \pm 112$ Da in the LNY daytime, respectively, compared to $415 \pm 93$ Da in the NYE daytime. It indicates that FW emissions contribute to the formation of HWM CHON compounds, particularly under daytime photooxidation. These newly formed compounds with low H/C and O/C ratios and high DBE values are likely oxidized aromatic compounds. As shown in Figure S4 and Table 3, nitrogen-containing compounds with AI > 0.5 were also more abundant in the LNY daytime (559 compounds) than the previous NYE nighttime (341 compounds).
CHON compounds were classified to $N_1O_3$–$N_1O_{14}$ and $N_2O_3$–$N_2O_{13}$ subgroups in all samples by the number of N and O atoms in their molecules (Figures 7 and S5). The total abundance of $N_1O_n$ subgroups was twice as much as that of $N_2O_n$ subgroups in each sample. After the FW period, with the reaction of photooxidation, each subgroup considerably increased in the LNY daytime, particularly for the $N_{1,2}O_{>7}$ subgroups, highlighting the importance of photooxidation to the formation of CHON compounds. As shown in Figure 8, the high intensity CHON compounds in samples were primarily with C
numbers of 15–25, O numbers of 2–8 and DBE values of 5–15, indicating that they potentially had one or more benzene rings in their molecules. However, there were numbers of CHON compounds with high carbon and oxygen content, and high unsaturation in LNY D sample. These unique compounds were likely HMW nitro-aromatic compounds.

Nitro-aromatic compounds are often observed in biomass burning aerosols (Iinuma et al., 2010;Kitanovski et al., 2012) and are potential contributors to light absorption as a component of brown carbon (Laskin et al., 2015;Lin et al., 2015;Desyaterik
et al., 2013). Although nitrogen-containing compounds did not increase significantly at NYE night, some biomass burning compounds might do. Figure 9 displays the ion intensity distributions of four nitro-aromatic compounds (i.e. $C_{10}H_7O_3N$, $C_{11}H_9O_3N$, $C_{12}H_{11}O_3N$, and $C_{16}H_{79}O_3N$) detected in biomass burning aerosols by Lin et al. (2015), which were just assigned by their molecular composition but not the chemical structure. Their intensities increased in the NYE N sample, particularly for $C_{11}H_9O_3N$ with its intensity being doubled. Nitro-aromatic compounds are produced in the atmosphere via the oxidation
of aromatic precursors in the presence of $NO_2$ (Laskin et al., 2015), and their relative yields increase with $NO_2/NO_3$ concentrations (Sato et al., 2007;Jang and Kamens, 2001), which can be released in large quantities during FW combustion processes. Moreover, high abundant PAHs from FW emission in the NYE can react more efficiently with $NO_2$ than their single-ring aromatic counterparts (Nishino et al., 2009).

## 3.5 CHOS compounds

More than one thousand CHOS compounds were assigned in the samples, accounting for 13−21% of all identified formulae. As shown in Figure 1, the relative intensities of CHOS compounds were the highest among four elemental compositional categories. FW emissions do influence the number and intensity of the CHO and CHON. However, CHOS increased

dramatically at NYE night, while they did not increase too much under sunlight in the following LNY daytime. As shown in Table 2, the number of CHOS compounds was 1146 in the NYE D sample, while it increased to 1979 during the FW event (NYE N). Moreover, during the FW event not only the number concentrations but also their intensities sharply increased to approximately twice as much as those before the FW event. Previous studies reported that a great deal of air pollutants released via FW burning in the NYE nighttime lead to a short-term pollution (Jiang et al., 2015;Tian et al., 2014). For example, higher concentrations of sulfate, $n$-alkanes ($C_{16-36}$), PAHs, and $n$-fatty acids ($C_{8-32}$) were observed in the FW burning night than the normal nights (Kong et al., 2015). These compounds might be the precursors of CHOS species (Riva et al., 2015;Riva et al., 2016;Passananti et al., 2016;Tao et al., 2014;Shang et al., 2016). The amount of sulfur in the firework was released into the air with the form of sulfur oxides during the combustion process, and further produced acidified sulfate seed aerosol, which considerably contributed to the formation of a large number of CHOS compounds via acid catalyzed reaction with biogenic and anthropogenic volatile organic compounds (VOCs) (Surratt et al., 2008;Riva et al., 2015). For instance, the CHOS compounds derived from monoterpenes and sesquiterpenes, such as limonene, $\alpha/\gamma$-terpinene and $\beta$-caryophyllene, were detected only under acidic or strongly acidic sulfate seed aerosol conditions (Surratt et al., 2008;Iinuma et al., 2007a;Iinuma et al., 2007b;Chan et al., 2011). Meanwhile, the high levels of nitrogen oxides emitted by FW burning can promote the formation of some CHOS compounds (Surratt et al., 2008).

Unlike the normal daytime and nighttime, there was a noticeable change in CHOS species affected by the FW event in the NYE nighttime (Figure 1). Figure 2 shows that CHOS species doubled in the NYE nighttime relative to the NYE daytime. On the contrary, both the number concentration and intensity were less at night than in the daytime in the normal day, which implies that FW plays an important role in the formation of CHOS compounds at night.

CHOS compounds were classified to $O_4S_1$–$O_{13}S_1$ subgroups by the number of O and S atoms in their molecules (Figure 10 and S6). Most of them had more than or equal to four O atoms of each S atom in their molecules, which supports the assignment of a sulfate group in the molecules. They are likely organosulfates (OSs), which is an important component of SOA formed by both daytime photooxidation and nighttime chemistry such as $NO_3$ oxidation. Unlike the CHO and CHON compounds, during the FW event, both the number and the intensity of each subgroup considerably increased in the NYE nighttime, not the LNY daytime, highlighting the importance of nighttime chemical oxidation to form CHOS compounds.

Table 2 demonstrates the arithmetic and weighted mean elemental ratios for each species of samples. The average molecular weights of CHOS compounds increased from 385±76 Da to 433±97 Da. Under the influence of firework emission, both the O/C and H/C ratios decreased in the NYE N aerosol, while the DBE and the DBE /C ratio increased. Formulae with $0 < H/C \leq 1.0$ and O/C $\leq 0.5$ dominantly have high DBE values ($\geq 7.0$), which is consistent with oxidized PAHs, e.g. the smallest PAH, naphthalene ($C_{10}H_8$) has an H/C of 0.8 and a DBE of 7 (Tong et al., 2016;Feng et al., 2012). Moreover, as shown in Figure 11 and Figure S7, in contrast to CHO and CHON species, numbers of CHOS compounds with high DBE ($\geq7.0$) were only detected in the NYE N aerosol; most of them fall into the area of AI > 0.5. The highest number of these PAH-like CHOS compounds was severally found in the NYE N sample with 125 ions, compared to only 68 ions in Normal N sample

(Table 3). These reflect that the FW emissions have an important influence on particle composition, especially for the aromatic-like compounds.

To further evaluate the characteristics of CHOS species producing during the FW burning, more than 92% of them were found to contain only one sulfur atom in each sample. Compounds that present a number of oxygen atoms greater than or equal to 4S (O ≥ 4S), potentially with a –$OSO_3H$ group, were tentatively regarded as OSs (Wang et al., 2016;Lin et al., 2012b). They considerably contribute to the yield of SOA (Tolocka and Turpin, 2012). However, tandem MS experiments were not conducted on the ions detected in samples. Hence, other sulfur-containing compounds, such as sulfonates, may also be involved due to the lack of using tandem MS experiments to provide insights into the exact structures (Riva et al., 2015;El Haddad et al., 2013).

The detailed composition coupled with molecular weights of OSs in the NYE D (1125 OSs) and NYE N (1945 OSs samples were displayed in Figure 11. Compared to the OSs in the NYE D sample, a dense distribution of one thousand compounds with high DBE (> 7) was found during the FW event, particularly for those within the high molecular weight (HMW; > 400 Da) region. Obviously, most of them had high $X_c$ (> 2.5, indicative of aromatics) and relatively low H/C (< 1.5) and O/C (< 0.5) ratios. Moreover, these highly unsaturated compounds had higher intensity, which indicated larger abundance of them in urban aerosols. These OSs with distinctive characteristics of high unsaturation were aromatic OSs, which were probably derived from aromatic VOCs or PAHs.

Furthermore, to illustrate the differences among the measured OSs, they are divided into three main classes. Group A includes aliphatic OSs with DBE ≤ 2, characterized by long alkyl carbon chains, which is highly saturated. Group B includes aromatic-like OSs, detected by the $X_c$ with the value of $X_c$ > 2.5, which has high degree of unsaturation. Group C includes the rest fraction except for groups A and B, which has a moderate degree of saturation and is similar to biogenic OSs. As shown in Figure 11, there were 322 aliphatic OSs and 125 biogenic OSs in NYE N sample, and 292 and 103 of those in NYE D sample, respectively. Moreover, the aliphatic OSs of $C_{12}H_{24}O_5S$, $C_{18}H_{36}O_6S$, and $C_{10}H_{16}O_9S$, and the biogenic OSs of $C_{10}H_{18}O_5S$ and $C_{10}H_{16}O_7S$, which were separately derived from alkanes and fatty acids (Riva et al., 2016;Passananti et al., 2016;Shang et al., 2016) and α/β-pinene (Surratt et al., 2008), and their corresponding family series ($C_nH_{2n}O_5S$, $C_nH_{2n}O_6S$, $C_nH_{2n-4}O_9S$, $C_{10}H_{2n-2}O_5S$ and $C_nH_{2n-4}O_7S$) were all detected in the aerosols. Nonetheless, the number of aromatic-like compounds (1498 OSs) increased dramatically in the NYE nighttime, particularly for the HMW compounds, compared to those (730 OSs) in the daytime. In addition, aromatic-like OSs, form not only in the daytime with photochemical reaction, but also in the nighttime via unknown formation pathways such as $N_2O_5$ oxidation. Plenty of aromatic-like ones rapidly formed in the NYE nighttime due to FW emissions, releasing plentiful aromatic VOCs as precursors of OSs. Riva et al. (2015) demonstrated the enhanced formation of OSs and sulfonates from both NAP and 2-MeNAP in the presence of acidified sulfate seed aerosol via comparison of side-by-side experiments. For instance, $C_9H_{10}O_5S$, $C_{10}H_{10}O_6S$, and $C_{10}H_{10}O_7S$, derived from 2-MeNAP (Riva et al., 2015), and their corresponding family series ($C_nH_{2n-8}O_5S$, $C_nH_{2n-10}O_6S$, and $C_nH_{2n-10}O_7S$) were detected in FW-related aerosols. These potentially explain that high abundance of aromatic-like OSs in

aerosols could be formed through the sulfate ion and PAHs emitted from FW burning at night without the presence of photoreaction.

## 4 Conclusion

We investigated HMW organic compounds in urban aerosols collected during the Chinese New Year in Beijing, including the periods of before FW (in the NYE daytime), during FW (in the NYE nighttime), and after FW (in the LNY daytime) by the usage of ESI FT-ICRMS. Three dominant categories of organic compounds, including CHO, CHON, and CHOS species, were measured and discussed. About 6,000 organic compounds were detected in the NYE daytime, while up to almost 7,000 in the NYE nighttime and 9500 in the LNY daytime. Moreover, the DBE values, the indicator of unsaturation, of them also clearly increased. Although they were increased by the effects of FW emissions, the three species compounds showed different behaviors. For the CHO species, there was no significant change for both the number and total intensities of them detected during the FW event, while they doubled with photooxidation in the LNY daytime, compared to the compounds before the FW event. Similarly, there was a similar trend for the CHON species as well as CHO groups. These phenomena indicated that photochemical reactions have a great influence on the formation of CHO and CHON compounds.

Sulfur-containing compounds increased dramatically at the NYE night. The number of CHOS specie was nearly twice in the NYE nighttime than those in the NYE daytime. About 92% of them were OSs, which rapidly increased in the NYE nighttime when large amount of pollutants emitted. High abundances of CHOS species with low H/C and O/C ratios and high DBE affected by FW emission are dominated by aromatic-like compounds such as aromatic carboxylic acids, nitro-aromatics, and poly-aromatic compounds; they are potentially contributors to atmospheric brown carbon that affects the physicochemical properties (e.g. light absorption and volatility) of atmospheric aerosols. Our results highlight that FW emission is a significant contributor to HMW organic aerosols, which needs to be considered in atmospheric chemical models for regional air quality.

*Data availability*. The dataset for this paper is available upon request from the corresponding author (fupingqing@tju.edu.cn).

*Competing interests*. The authors declare that they have no conflict of interest.

*Acknowledgements*. This work was supported the National Natural Science Foundation of China (Grant Nos. 41625014, 41961130384, and 41571130024).

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

**Table 1.** The concentrations of chemical components in the Beijing aerosol samples.

| Sample ID | Sampling Date | OC ($\mu g\ m^{-3}$) | EC ($\mu g\ m^{-3}$) | WSOC ($\mu gC\ m^{-3}$) | WSON ($\mu gN\ m^{-3}$) | ΣPAHs* ($ng\ m^{-3}$) | Water-soluble ion concentrations ($\mu g\ m^{-3}$) | | | | | | |
|---|---|---|---|---|---|---|---|---|---|---|---|---|---|
| | | | | | | | $SO_4^{2-}$ | $NO_3^-$ | $Cl^-$ | $NH_4^+$ | $K^+$ | $Mg^{2+}$ | $Ca^{2+}$ |
| Normal D | 21st Jan | 11.9 | 3.93 | 2.81 | 0.07 | 39.2 | 5.18 | 2.27 | 1.14 | 0.71 | 0.51 | 0.23 | 1.60 |
| Normal N | 21st Jan | 9.27 | 2.32 | 2.70 | 0.12 | 71.7 | 4.65 | 2.03 | 1.56 | 0.61 | 0.65 | 0.16 | 0.67 |
| NYE D | 22nd Jan | 13.0 | 3.22 | 5.10 | 0.36 | 44.4 | 5.66 | 4.19 | 2.49 | 2.00 | 1.90 | 0.24 | 1.38 |
| NYE N | 22nd Jan | 23.3 | 8.65 | 11.6 | 1.36 | 165.7 | 65.4 | 5.34 | 55.2 | 0.17 | 102 | 2.92 | 2.25 |
| LNY D | 23th Jan | 11.2 | 3.20 | 3.83 | 0.16 | 29.7 | 6.43 | 3.16 | 3.54 | 0.95 | 3.99 | 0.53 | 1.39 |
| LNY N | 23th Jan | 14.1 | 4.73 | 5.14 | 0.58 | 50.9 | 11.6 | 3.31 | 6.49 | 0.47 | 11.0 | 0.83 | 1.13 |

*: ΣPAHs; The total concentration of eighteen detected PAHs. NYE: new year eve (detailed days). LNY: lunar New Year's Day (detailed days).

**Table 2.** The number of compounds in each subgroup and arithmetic and weighted mean elemental ratio for each subgroup in NYE D and NYE N samples.

| Parameters | NYE D | | | | NYE N | | | |
|---|---|---|---|---|---|---|---|---|
| | All compounds | CHO | CHON | CHOS | All compounds | CHO | CHON | CHOS |
| Number frequency | 5854 | 2045 | 2623 | 1146 | 6836 | 2260 | 2597 | 1979 |
| Molecular weight (Da) | 405±89 | 407±100 | 415±93 | 385±76 | 439±99 | 424±107 | 445±100 | 433±97 |
| O/C | 0.38±0.14 | 0.33±0.11 | 0.36±0.12 | 0.44±0.15 | 0.37±0.13 | 0.30±0.12 | 0.33±0.12 | 0.37±0.14 |
| O/C$_w$ | 0.38 | 0.33 | 0.36 | 0.41 | 0.37 | 0.30 | 0.33 | 0.37 |
| H/C | 1.23±0.36 | 1.14±0.37 | 1.13±0.32 | 1.49±0.42 | 1.23±0.37 | 1.18±0.37 | 1.11±0.28 | 1.35±0.42 |
| H/C$_w$ | 1.25 | 1.10 | 1.11 | 1.63 | 1.24 | 1.16 | 1.08 | 1.40 |
| OM/OC | 1.70±0.22 | 1.53±0.15 | 1.65±0.16 | 1.87±0.23 | 1.70±0.23 | 1.50±0.16 | 1.61±0.17 | 1.75±0.22 |
| OM/OC$_w$ | 1.70 | 1.53 | 1.64 | 1.84 | 1.71 | 1.50 | 1.60 | 1.76 |
| DBE | 9.35±4.01 | 10.7±5.00 | 11.0±4.36 | 5.53±3.85 | 10.1±4.82 | 10.7±5.29 | 12.1±4.43 | 8.03±5.19 |
| DBE$_w$ | 8.94 | 10.5 | 11.0 | 4.21 | 9.52 | 10.5 | 12.1 | 6.97 |
| DBE/C | 0.45±0.18 | 0.48±0.18 | 0.52±0.16 | 0.32±0.18 | 0.45±0.17 | 0.45±0.18 | 0.52±0.14 | 0.37±0.21 |
| DBE/C$_w$ | 0.45 | 0.50 | 0.53 | 0.25 | 0.45 | 0.46 | 0.53 | 0.36 |

**Table 3.** The number of compounds with various AI values.

| Sample ID | Parameter | CHO | CHON | CHOS |
|---|---|---|---|---|
| NYE D | AI=0 | 86 | 175 | 506 |
| | 0<AI<0.5 | 1460 | 2084 | 608 |
| | 0.5≤AI<0.67 | 457 | 361 | 32 |
| | 0.67≤AI | 42 | 3 | 0 |
| NYE N | AI=0 | 88 | 78 | 583 |
| | 0<AI<0.5 | 1686 | 2175 | 1271 |
| | 0.5≤AI<0.67 | 426 | 340 | 120 |
| | 0.67≤AI | 58 | 1 | 5 |
| LNY D | AI=0 | 124 | 91 | 401 |
| | 0<AI<0.5 | 2219 | 2954 | 772 |
| | 0.5≤AI<0.67 | 692 | 545 | 75 |
| | 0.67≤AI | 85 | 14 | 0 |
| LNY N | AI=0 | 127 | 160 | 489 |
| | 0<AI<0.5 | 1926 | 1931 | 1050 |
| | 0.5≤AI<0.67 | 512 | 418 | 85 |
| | 0.67≤AI | 53 | 6 | 2 |
| Normal D | AI=0 | 54 | 105 | 558 |
| | 0<AI<0.5 | 1629 | 1970 | 760 |
| | 0.5≤AI<0.67 | 440 | 296 | 80 |
| | 0.67≤AI | 45 | 7 | 1 |
| Normal N | AI=0 | 79 | 96 | 442 |
| | 0<AI<0.5 | 1545 | 1642 | 733 |
| | 0.5≤AI<0.67 | 413 | 402 | 68 |
| | 0.67≤AI | 34 | 0 | 0 |

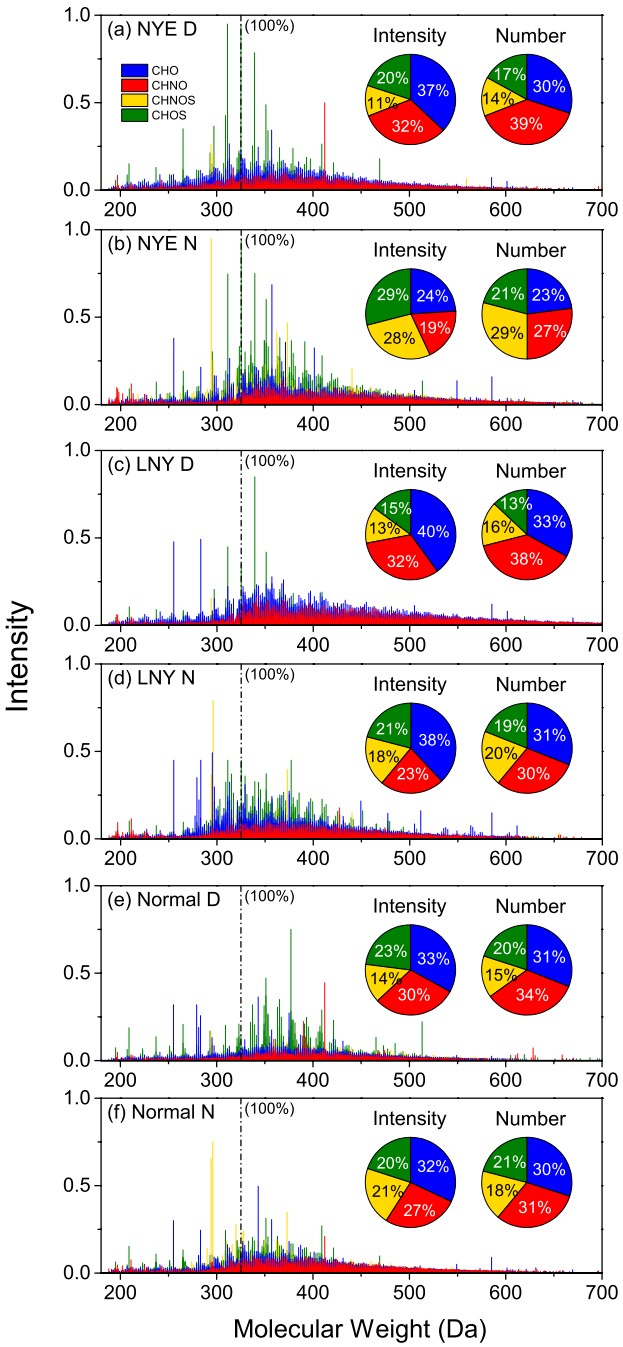

**Figure 1.** Distribution of relative intensity, number and intensity fractions of CHO, CHNO, CHOS and CHNOS compound in WSOC isolated from aerosol samples detected in FT-ICRMS. The detailed molecular characteristics of CHNOS is discussed in another study.

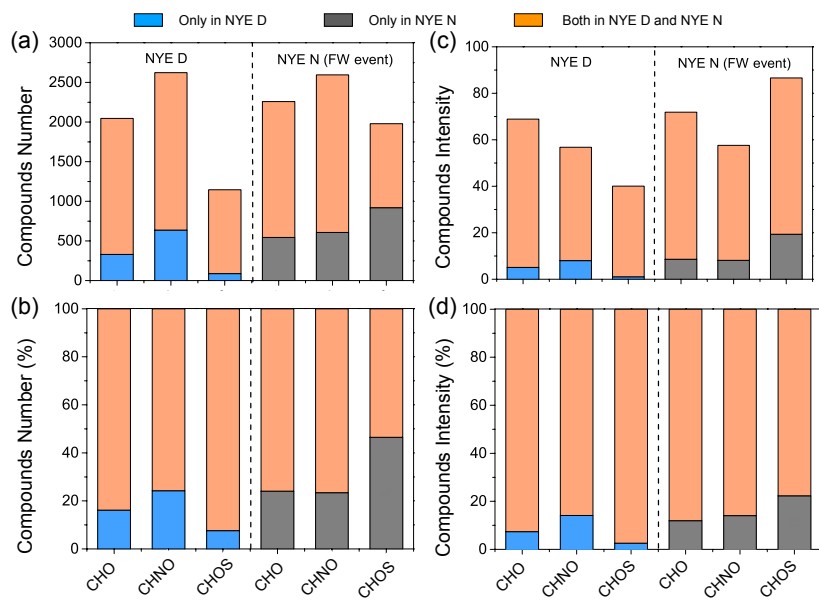

**Figure 2.** The number (**a, b**) and intensity (**c, d**) of molecular formulae associated with three categories compounds in NYE D (before the FW event) and NYE N (during the FW event) samples, and common formulae present at both samples.

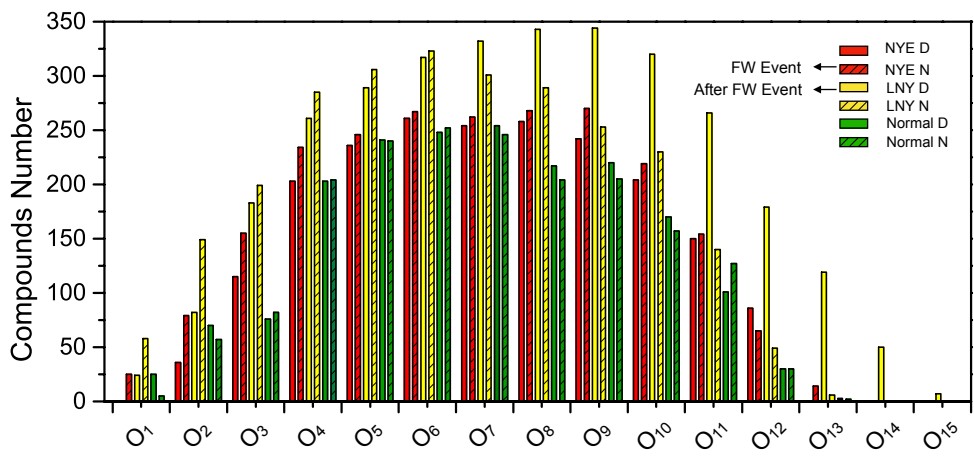

**Figure 3.** Classification of CHO species into subgroups according to the number of O atom in their molecules.

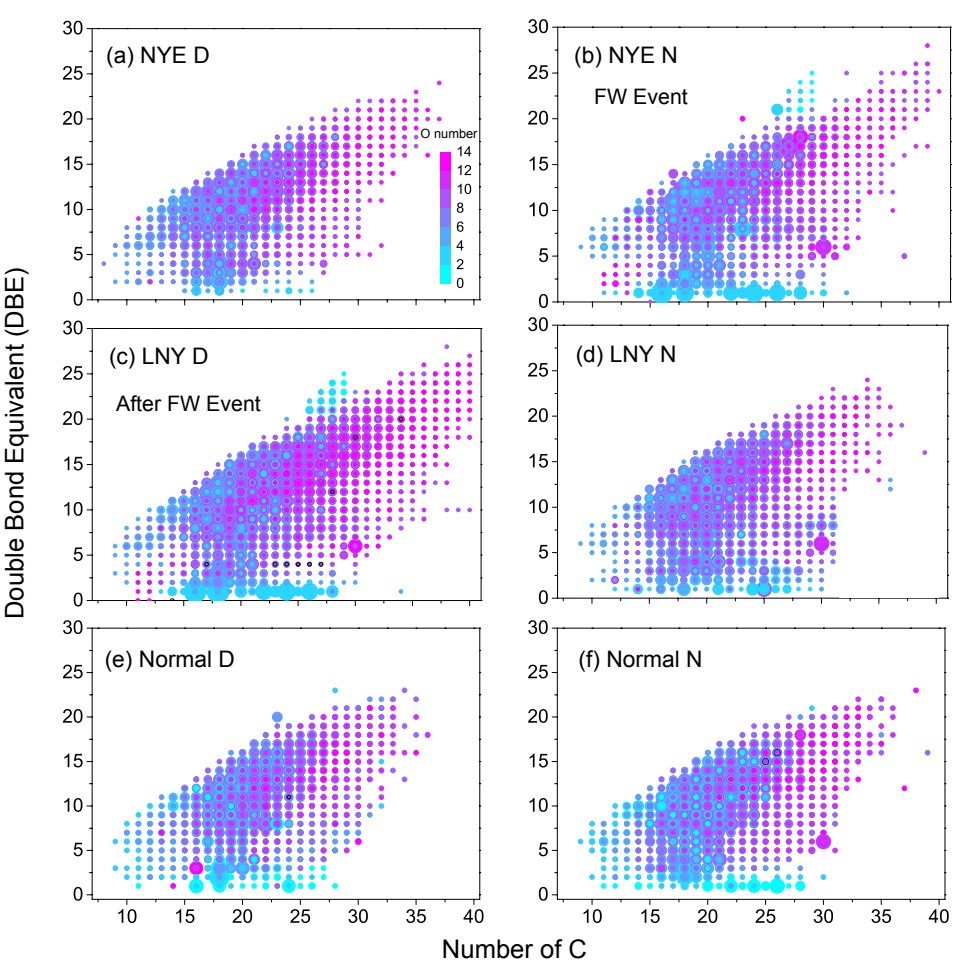

**Figure 4.** Double bond equivalent (DBE) vs. number of C atoms for CHO species. The color bar denotes the number of O atoms. The size of the symbols reflects the relative peak intensities of molecular formulae on a logarithmic scale.

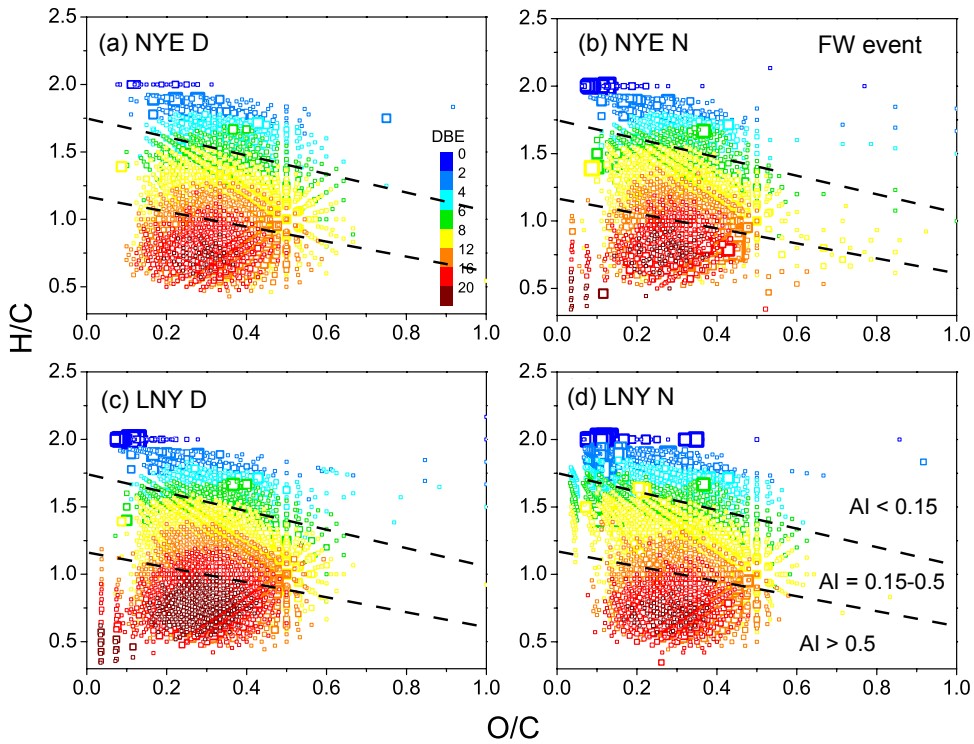

**Figure 5.** Van Krevelen diagrams (the H/C via O/C ratios) for the CHO compounds with various aromatic index (AI) values ranges. The dashes lines separate the different AI regions. The size of the symbols reflects the relative peak intensities of compounds on a logarithmic scale.

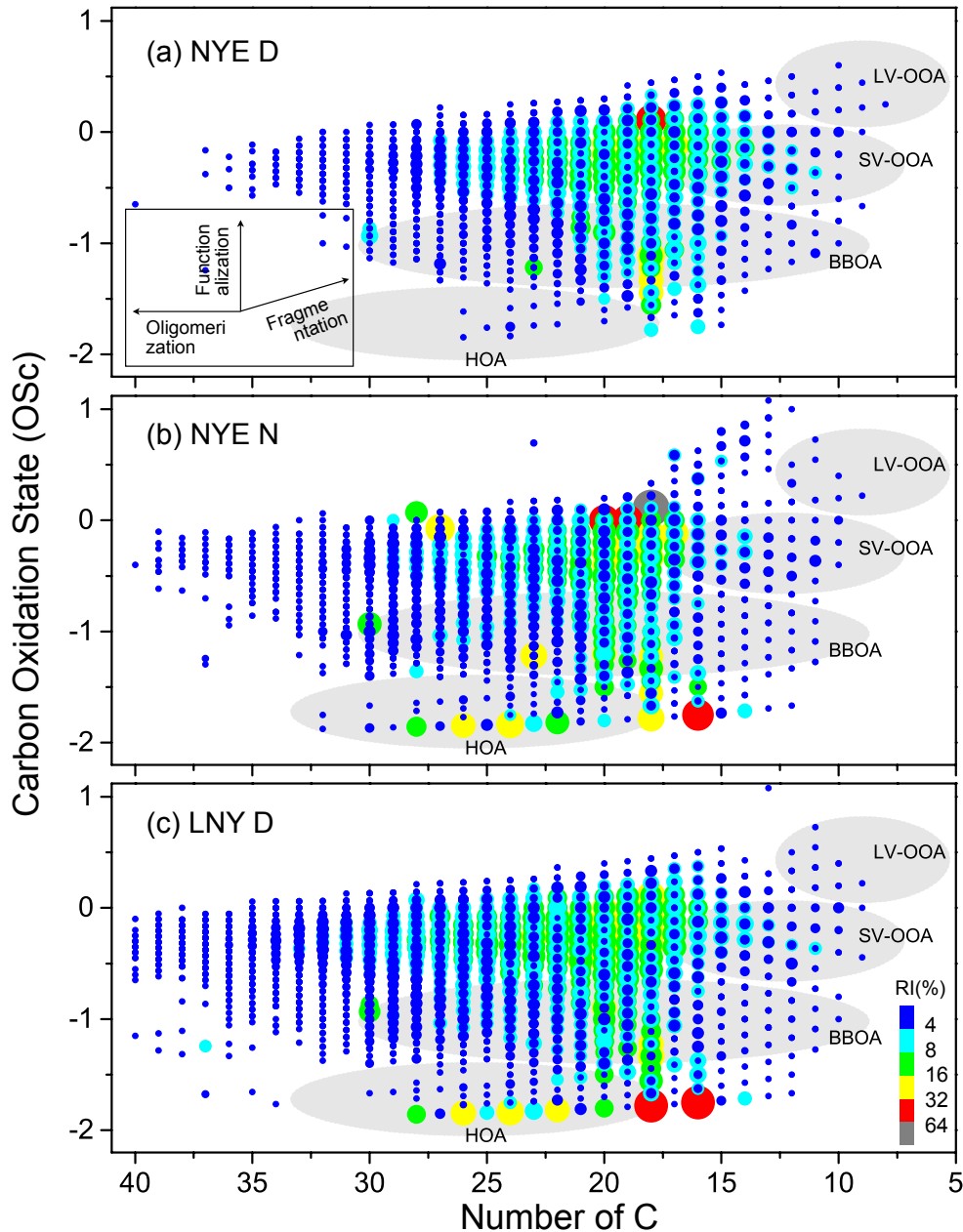

**Figure 6.** Overlaid carbon oxidation state (OS$_C$) symbols for CHO compounds in NYE D (a), NYE N (b), and LNY D (c) samples. The size and color bar of the markers reflects the relative peak intensities of compounds on a logarithmic scale. The gray areas were marked as SV-OOA (semi-volatile oxidized organic aerosol), LV-OOA (low-volatility oxidized organic aerosol), BBOA (biomass burning organic aerosol) and HOA (hydrocarbon-like organic aerosol) (Kourtchev et al., 2016;Kroll et al., 2011).

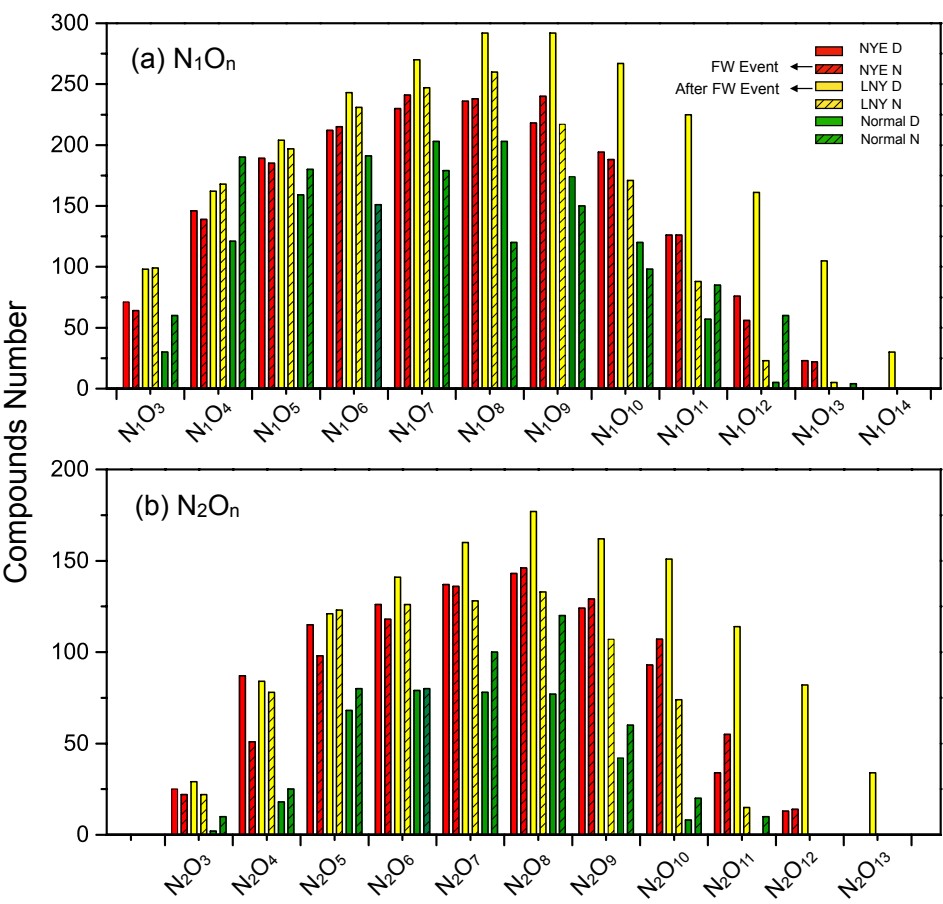

**Figure 7.** Classification of CHNO species into subgroups according to the number of N and O atoms in their molecules.

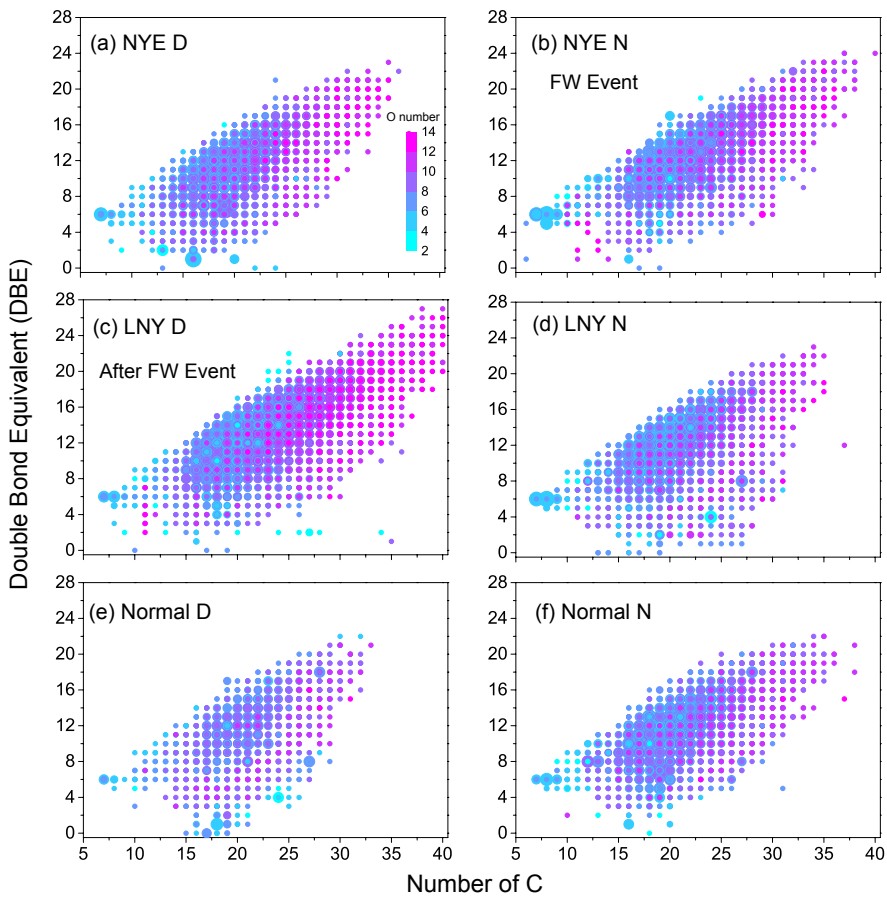

**Figure 8.** Double bond equivalent (DBE) vs. number of C atoms for CHNO species. The color bar denotes the number of O atoms. The size of the symbols reflects the relative peak intensities of molecular formulae on a logarithmic scale.

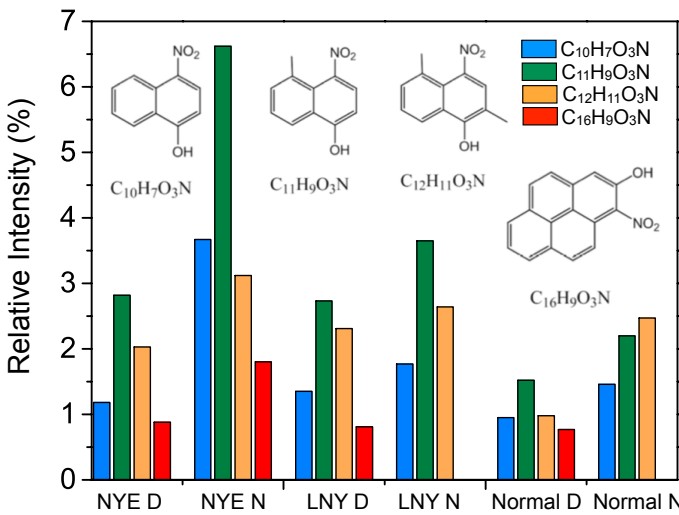

**Figure 9.** Ion intensity distributions of selected tentatively identified compounds in individual samples. They may be nitro-aromatics and their potential structures have been reported by Lin et al. (2015).

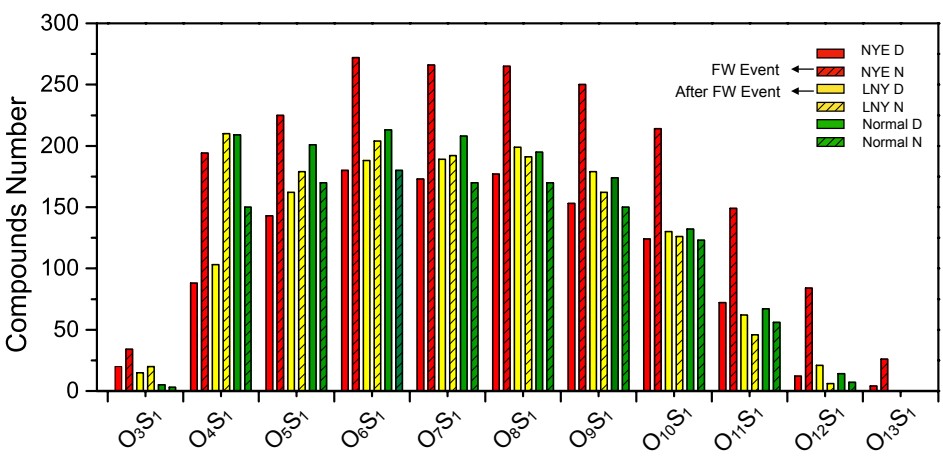

**Figure 10.** Classification of CHOS species into subgroups according to the number of O and S atoms in their molecules.

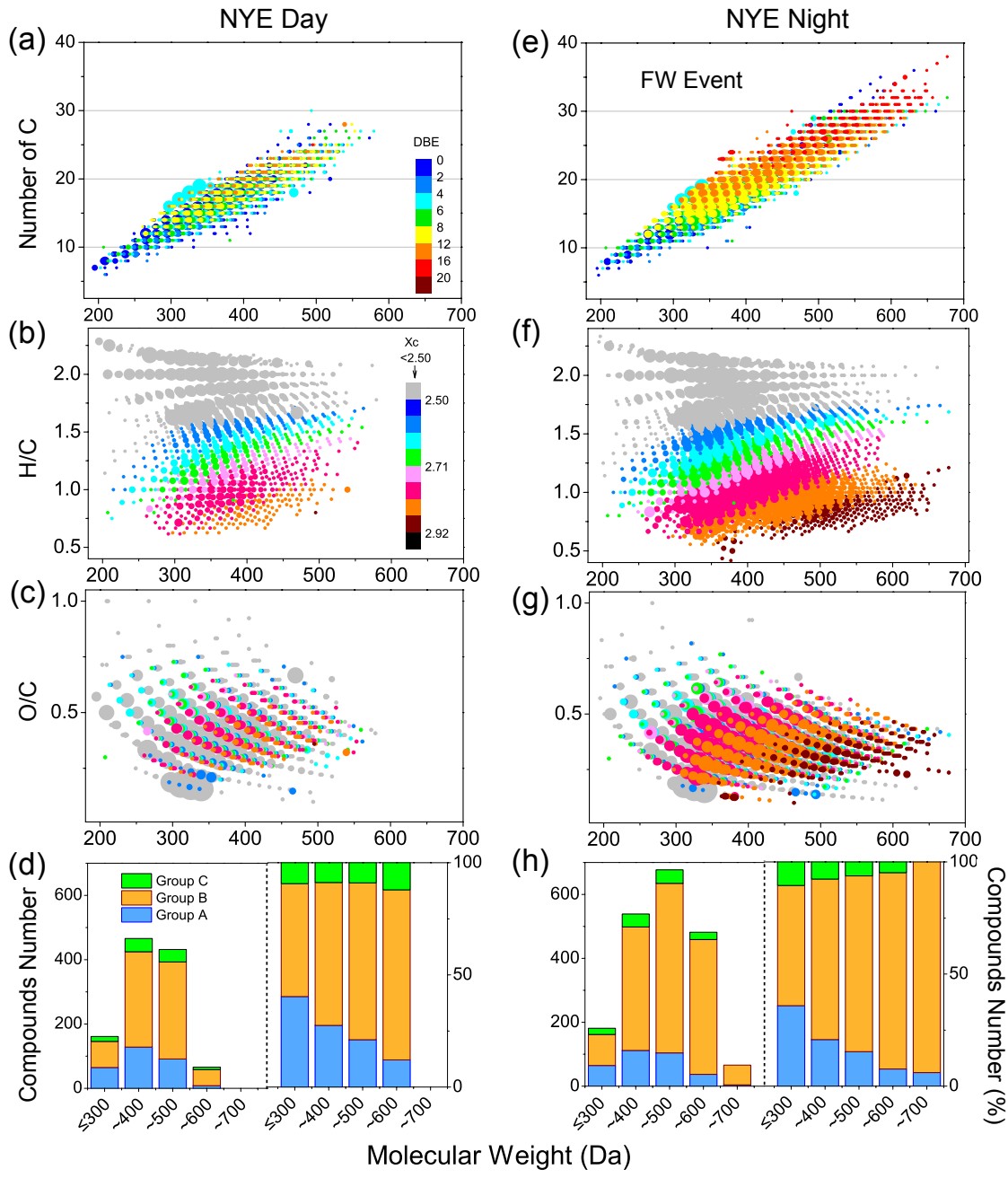

**Figure 11.** The carbon chain length (**a, e**), H/C (**b, f**) and O/C (**c, g**) ratios, different groups (**d, h**) distributions via molecular weights of OSs in NYE D (before the FW event) and NYE N (during the FW event) aerosols. Group A includes the aliphatic OSs with DBE≤2; Group B includes the aromatic-like OSs with $X_c$>2.5; Group C includes the biogenic OSs. The color bar in (**a, e**) and (**b, c, f, g**) denotes the number of DBE and the value of $X_c$.