# Peer review of "Molecular Characterization of Firework-Related Urban Aerosols using FT-ICR Mass Spectrometry"

_Atmospheric Chemistry and Physics, 2019_

## Referee Comment (RC1) · Anonymous Referee #1 · 23 Feb 2020

General comments.

The manuscript presents an analysis of 6 samples collected on the days before, during, and after a major fireworks emission source in Beijing. The authors used FT-ICR MS to characterize the CHO, CHON, and CHOS compounds identified in the samples. The data analysis is thorough and the data sets and results are presented in a clear format. However, there are multiple places where additional information should be included and clarifications given. I recommend this for publication in ACP after the following specific comments are addressed.

Specific Comments

1. The CHO, CHON, and CHOS compounds are the only ones discussed here. There are clearly CHONS compounds present in the sample and the caption on Figure 1 says that these compounds "were discussed in other study". If this study is published, please provide the citation. Also, please include this information somewhere in the text in addition to the caption. The end of the introduction would be a good location.

2. In the abstract and in the conclusions a reference is made to brown carbon molecules. Were any UV-vis measurements made that would support the idea that some of these molecules can absorb visible radiation? If not, was there any observation (by eye) that some of these extracts were more brown?

3. In the abstract, the statement "the co-variation of CHO, CHON, and CHOS subgroups was suggested to be associated with multiple atmospheric aging process of aerosols including the multiphase redox chemistry driven by NOx, O3, and OH." This sentence is a little confusing, what co-variation is being referred to here?

4. For the experimental, was the possible presence of phosphorous included in the assignment? If not, why?

5. For the FT-ICR, what mass was the instrument tuned too and what is the lower mass cut-off for the ion trap?

6. A file of peak lists and assignments for all the samples would be very helpful for scientists wishing to build on this work. Can this be included as additional supplemental files? What percentage of the identified peaks were assigned? What fraction of the total signal does this correspond to?

7. How was the signal from the field blank handled? Were peaks that were found in the blank excluded? Or was the S/N relative to the blank used?

8. On page 6 lines 13-14 you state "the peak intensities of the ions could be compared by assuming that matrix effects were relatively constant". Please clarify that this is a sample to sample comparison and not that ion intensities for different compounds

within a sample were compared. Those will be affected by ionization efficiency (as you state).

9. On page 7 lines 16-17 you state: " Moreover, the number and total intensities...(Figure 2)." I am confused what comparison is being made here since this paragraph is about CHO compounds, please clarify.

10. The carbon oxidation state discussion and figure have multiple areas of modification:

a. Figure 6 is very hard to read, even with color. I cannot see the blue markers (NYE D) under all the others and especially when they are on top of the green ovals.

b. In the text, it sounds like the authors are saying that compounds with molecular formula that overlap with different green ovals (BBOA, SV-OOA, etc.) correspond to those compound types. Specifically I recommend adjusting the text that starts on line 31 page 8 to clarify that these groupings are for previous measurements of ambient aerosol samples. The phrasing "molecules with OSc between xx and xx with carbon atoms more than 7 are associated with xxx" sounds like the molecules in this study are being assigned to these groups. If this is the intended interpretation, please see my caution in comment 16.

11. On page 9 lines 12-14, the authors state that the molecular weight increased during the FW events for the CHON compounds. However, all these numbers are within the reported error of each other.

12. The paragraph on page 9 starting on line 18 is confusing. Which type of oxidized nitrogen group is being assigned for which sample? Both organonitrates and nitro-aromatics are discussed but it is unclear if these are for different samples.

13. The trends shown in Figure 9 are interesting and the caption is appropriately clear on how tentative these assignments are. The text that corresponds to this figure (page 9, lines 28-34) should also be adjusted to indicate that these are not structural

assignments.

14. For the CHOS compounds, is there any reason that some of these could not be primary emissions? Have any FT-ICR studies been carried out with samples collected closer to the FW source?

15. I recommend changing the label for the sulfur section from OSs to CHOS. This will match the rest of the paper, it will decrease confusion with OSc, and will be better given that no MS/MS studies were done to positively identify them as organosulfates (as stated on page 11).

16. On page 12 line 12-13, the authors state: "Moreover, a great part of the FW affected ions with high intensity were potentially the BBOA". What data is this conclusion being drawn from? Is this coming from the oxidation state figure/analysis? If it is coming from the oxidation state figure, I urge caution with this type of conclusion. The carbon oxidation state is a great metric for analyzing atmospheric aging, but molecules from different sources can have similar carbon oxidation state and carbon number ranges. Please also remember that the analysis here is only looking at material that was bound and then eluted on the SPE column, is water soluble, and is easily ionized in negative ion mode. Caution should be used when making aerosol source identifications from the molecular formulas found here to ones found for different sample types with different preparation steps.

---

## Referee Comment (RC2) · Anonymous Referee #2 · 3 Mar 2020

Xi et al. propose a study on the characterization of ambient aerosols using an FT-ICR. 6 samples were analyzed and compared to evaluate the impact of firework on air quality. Overall the data reported in this study are coherent and the structure of the paper is clear. However, additional information should be added as well as some explanation to make this paper more comprehensive.

Page 2, lines 8-9: please reformulate.

Page 2, line 31: Please provide more information regarding the sampling of the aerosol: the size of the particles; i.e., PM1, 2.5, 10? high-volume samplers?

Page 3, line 2-3: I recommend the authors to use a simpler naming system. e.g.,

before-FW-1, after-FW-2, during-FW-1,... it would be much easier to follow the discussion.

Page 4, line 1: The authors should explain why these samples were analyzed only in negative mode.

Page 4, section 2.4: - why did the authors choose an S/N> 6, which is more restricting? Why not using an S/N ratio > 3, which is commonly defined as LoD?

- Is the peak assignment perform before or after blank subtraction? While the authors mentioned that blank filters were collected and analyzed, no information is providing regarding how the blank samples were used for the data analyzing.

- Why did the authors start at m/z 185 rather than m/z 100 as mentioned earlier in the manuscript? A significant amount of potential OA compounds can be missing.

- The authors should not claim any semi-quantitative results as the sensitivity of the ESI is extremely dependent on the functional groups of individual compounds as reported in many studies.

Page 5, lines 26-28: The authors should provide either some references or some supporting information to support their statement.

Page 5, lines 29: How do the authors know the distribution of the ions/compounds as a function of the size provide either reference or supporting information.

Page 6, lines 2-4: While the concentrations of $SO_4^{2-}$, $Cl^-$ and $K^+$ are significantly different, this is not really the case for WSOC: there is an increase $\sim$ a factor of 2, but what's the daily variability? It is hard to conclude that FW produces a sharp increase. In other words, is the increase statistically different?

Page 6, line 12: How do the authors know the ionization efficiency of the observed compounds? i.e., the concentrations of some ions can be very high but with very poor ionization efficiency. This statement is purely speculative

Page 6, lines 13-14: That's an incorrect assumption/statement. The matrix effect is one aspect. I strongly encourage the authors to check basic studies on ESI and revise their manuscript. Indeed the peak intensity can be impacted by the matrix but also depends on the ionization efficiency of individual compounds which is based on the functional groups present in each compound.

Page 6, line 19: This is actually surprising and not consistent with "normal" product distribution. Indeed in most of the previous studies, most of the identified ions are between 150-250 (i.e., monomers type) and a second mode is present between 300-400 (dimers or high molecular weight compounds), see the previous characterization using ESI-MS (QTOF, Orbitrap, and FT-ICR). The authors should comment on such a curious product distribution. The authors should clearly mention that probably the vast majority of the compounds were lost during the sampling preparation.

What is the reason to remove a major fraction of the organic compounds during the sampling preparation (i.e., page 3, lines 26-27)?

Page 6, line 33 and page 7, lines 1-2: This is overstated, the authors should provide deeper statistical analysis before making such a statement. Are the numbers really statistically different?

Page 7, line 8: Please provide numbers to support such a statement.

Page 7, lines 12-14: I am confused by this sentence. Why did the authors refer to Ms2 studies while they didn't perform such an analysis.

Page 7, line 30: This statement is incorrect. It is not in urban aerosol but in the filter extract, i.e., after removing a major fraction of organic aerosol components.

Page 9, lines 20-27: This section is confusing. please clarify, i.e., within the same paragraph the authors claim that CHON with O>3 are likely organonitrates and a few lines below nitroaromatics. In addition, the authors should keep in mind the difference in terms of ionization efficiency of such compounds: i.e., compounds containing nitrofunctional groups have a very high ionization efficiency (e.g., nitroaromatics), unlike organonitrate compounds.

Why is the CHONS group not discussed in the paper?

---

## Referee Comment (RC3) · Anonymous Referee #3 · 5 Mar 2020

General Comments:

The article studied the characteristic of CHO, CHNO, and CHOS before, during, and after FW event. Many species were detected by FT-ICR MS, and were analyzed through the manuscript. Furthermore, potential sources of these subgroups were also discussed through many calculations. There are many data and analysis methods which help the reader to understand the different pollution characters of aerosols during six periods. The whole article was aimed to discuss the event of firework-related urban aerosols before, during, and after New Year's Eve evening. The author just discussed the redox chemistry driven by NOx, O3, and OH, but the impact of combustion processes during the FW event wasn't discussed. What the relation between combustion processes and the three subgroups? I think the combustion processes is an important factor for the pollution during FW event. For example, there are amount of sulfur in the firework which many contribute the formation of CHOS species. Besides, the meteorology parameters were not contained in the article, which is hardly to analysis the sources of these subgroups studied in the article. for example, the author indicated the "multiphase" redox chemistry is important for the detected species formation, but how about the RH during these periods? The article should be revised according the comments and then can be published.

Specific Comments:

Introduction: Why the author studied subgroups of CHO, CHNO, and CHOS during these periods?

Page 5 Line 28-30: The author ascribed the increase of Mg2+ and Ca2+ to the dust particles increased were not exactly. Mg would exist in the FW. Did the author get the PM10 data?

Page 6 Line 20: CHNOS was mentioned here and also in Figure 1. Why the author didn't discuss CHNOS? What the relationship between CHNOS with CHNO and CHOS?

Page 7 Line 8: while the relative abundance of "four" categories compounds: "four" or "three"?

Page 8 Line 18: Xc can help to more precisely identify and characterize aromatic and condensed aromatic compounds in highly complex WSOC mixtures, why AI method was used in the manuscript?

Figure 6: I can't understand this picture, the markers can't be seen clearly, the green areas can't be understood.

Page 10 Line21-24, Page 11 Line 25: The author highlights the importance of nighttime

chemical oxidation to the formation of CHOS compounds, what was the evidence? How the combustion process impacted the formation of CHOS during FW event?

---

## Author Comment (AC1) · 27 Apr 2020

**Authors' Responses to Reviewer #1**

We appreciate the detailed and constructive comments and suggestions from the reviewer. The point-to-point responses to the comments are listed as below.

The *Reviewer comments are black italic font* and the Author responses are blue font.

*General comments.*

*The manuscript presents an analysis of 6 samples collected on the days before, during, and after a major fireworks emission source in Beijing. The authors used FT-ICR MS to characterize the CHO, CHON, and CHOS compounds identified in the samples. The data analysis is thorough and the data sets and results are presented in a clear format. However, there are multiple places where additional information should be included and clarifications given. I recommend this for publication in ACP after the following specific comments are addressed.*

**Response:** We appreciate the valuable comments from the reviewer. We have made changes/modifications to both the main text and the supplemental information. Detailed responses are shown below.

*Specific Comments*

*1. The CHO, CHON, and CHOS compounds are the only ones discussed here. There are clearly CHONS compounds present in the sample and the caption on Figure 1 says that these compounds "were discussed in other study". If this study is published, please provide the citation. Also, please include this information somewhere in the text in addition to the caption. The end of the introduction would be a good location.*

**Response:** Thank you. CHONS compounds are obviously subject to nighttime chemical oxidation. In the present study, we investigated the molecular characteristics of CHO, CHON, and CHOS species in a detailed way. Due to the limited length of the paper, the characteristics of CHONS and their volatility estimated by using some molecular corridor method will be discussed in a detailed and thorough way in another manuscript that is under preparation. We have added the following sentence to the manuscript in the end of the introduction in addition to the caption (page 3 lines 8-9):

"In addition, the detailed molecular characteristics of CHNOS species and their volatility using

a molecular corridor method will be present in another study."

*2. In the abstract and in the conclusions a reference is made to brown carbon molecules. Were any UV-vis measurements made that would support the idea that some of these molecules can absorb visible radiation? If not, was there any observation (by eye) that some of these extracts were more brown?*

**Response:** Because of the limitations of measurement techniques and the lack of authentic standards of OSs that are detected in FT-ICRMS, we have not been able to make the UV-vis measurements of each compound so far. The conclusions resulted from the molecular composition, such as nitro-aromatic compounds, which are potential contributors to light absorption as a component of brown carbon.

*3. In the abstract, the statement "the co-variation of CHO, CHON, and CHOS subgroups was suggested to be associated with multiple atmospheric aging process of aerosols including the multiphase redox chemistry driven by NOx, O₃, and OH." This sentence is a little confusing, what co-variation is being referred to here?*

**Response:** The co-variation is referred to the previous sentence "…the number concentration of sulfur-containing compounds especially the organosulfates was increased dramatically by the FW event, whereas the number concentration of CHO and CHON doubled after the event." The number concentration of CHOS compounds was increased dramatically by the FW event at night, and they are mainly subject to nighttime chemical oxidation, e.g., $NO_3$, $N_2O_5$. But the number concentration of CHO and CHON doubled after the event during the day with photochemistry. The co-variation of CHO, CHON, and CHOS subgroups in the manuscript is referred to the same variation and different variation after being affected by fireworks emissions, and they are associated with the multiphase redox chemistry driven by NOx, O₃, and OH. To avoid confusing, we have changed the statement: (page 1 lines 27-28)
"…, which were associated with multiple atmospheric aging processes including the multiphase redox chemistry driven by $NO_x$, $O_3$, and $^{\bullet}OH$."

*4. For the experimental, was the possible presence of phosphorous included in the assignment? If not, why?*

**Response:** Phosphorus-containing compounds existed in our samples, but the number and

intensity of them only accounted for less than 0.01% of that in all assigned compounds. We assume that organophosphates should be more abundant in summertime aerosol samples from biogenic origin, not the wintertime ones. Therefore, phosphorus-containing compounds were not included in the discussion of our manuscript that mainly focuses on firework-influenced N- and S-containing aerosols.

*5. For the FT-ICR, what mass was the instrument tuned too and what is the lower mass cut-off for the ion trap?*

**Response:** For the FT-ICR MS analysis, the ion peaks of sodium formate complex ranging from approximately 180 Da to 1200 Da were tuned with an error of less than 1 ppm. And the lower mass cut-off for the ion trap was 100 Da.

*6. A file of peak lists and assignments for all the samples would be very helpful for scientists wishing to build on this work. Can this be included as additional supplemental files? What percentage of the identified peaks were assigned? What fraction of the total signal does this correspond to?*

**Response:** We thank the reviewer for this comment. As the data of the work also involves some unpublished articles, our dataset will be available upon request by the readers. There were approximately 67%~71% of the identified peaks were assigned in our samples. And the intensity of them accounted for approximately 70%~75% of the total signal. The information has been added in the manuscript. (page 4 lines 30-31)

"There were approximately 67−71% of the identified peaks were assigned in our samples. The intensity of them accounted for 70−75% of the total signal."

*7. How was the signal from the field blank handled? Were peaks that were found in the blank excluded? Or was the S/N relative to the blank used?*

**Response:** The assignments of the blank sample with a signal-to-noise ratio three times that of the aerosol sample were removed in our samples. The information has been added in the manuscript. (page 4 line 31- page 5 line 1)

"The molecular formulas in blank filters with a signal-to-noise ratio greater than that of the aerosol samples were subtracted from the real aerosol samples."

*8. On page 6 lines 13-14 you state "the peak intensities of the ions could be compared by assuming that matrix effects were relatively constant". Please clarify that this is a sample to sample comparison and not that ion intensities for different compounds within a sample were compared. Those will be affected by ionization efficiency (as you state).*

**Response:** Thank you very much for this suggestion. To clarify, we have changed the presentation in the manuscript (page 6 lines 32 - page 7 line 2):

"the peak intensities of the same ions could be compared among different samples by assuming that matrix effects were relatively constant."

*9. On page 7 lines 16-17 you state: "Moreover, the number and total intensities...(Figure 2)." I am confused what comparison is being made here since this paragraph is about CHO compounds, please clarify.*

**Response:** Thank you. The comparison is made for the unique CHO compounds (except for common compounds in two samples) between that only detected in NYE N sample and NYE D sample. To be more accurate and avoid any confusion, we have modified the statements in the manuscript (on page 8 lines 4-6):

"Except for common compounds in two samples, the number and total intensities of the unique compounds in NYE N sample (591 compounds) were slightly increased compared with those only in NYE D sample (376 compounds) (Figure 2)."

*10. The carbon oxidation state discussion and figure have multiple areas of modification:*
*a. Figure 6 is very hard to read, even with color. I cannot see the blue markers (NYE D) under all the others and especially when they are on top of the green ovals.*

**Response:** We thank the reviewer for this comment. To make it easier to read, we have modified the diagram as presented below. Due to the large amount of data, we presented each sample separately in one chart avoid overlapping. The explanatory graph was added with the lower left corner of the graph (a). The green ovals areas have been replaced by the gray areas.

[Figure]

**Figure 6.** Overlaid carbon oxidation state (OS_C) symbols for CHO compounds in NYE D (a), NYE N (b), and LNY D (c) samples. The size and color bar of the markers reflects the relative peak intensities of compounds on a logarithmic scale. The gray areas were marked as SV-OOA (semi-volatile oxidized organic aerosol), LV-OOA (low-volatility oxidized organic aerosol), BBOA (biomass burning organic aerosol) and HOA (hydrocarbon-like organic aerosol) (Kourtchev et al., 2016;Kroll et al., 2011)."

*b. In the text, it sounds like the authors are saying that compounds with molecular formula that overlap with different green ovals (BBOA, SV-OOA, etc.) correspond to those compound types. Specifically I recommend adjusting the text that starts on line 31 page 8 to clarify that these*

*groupings are for previous measurements of ambient aerosol samples. The phrasing "molecules with OSc between xx and xx with carbon atoms more than 7 are associated with xxx" sounds like the molecules in this study are being assigned to these groups. If this is the intended interpretation, please see my caution in comment 16.*

**Response:** Thank you. To clarify, we have adjusted the statement about the interpretation of different compound types (SV-OOA, LV-OOA, BBOA, and HOA) to:

"As shown in Figure 6, different OSC value and C number indicate different types of compounds as previously characterized by Kroll et al. (2011). The semi-volatile and low-volatility oxidized organic aerosol (SV-OOA and LV-OOA) have the values of OSC between −1 and +1 and carbon atoms less than 13, which are associated with that are produced by multistep oxidation reactions. The biomass burning organic aerosol (BBOA) has lower OSC, with OSC between −0.5 and −1.5 and carbon atoms more than 7. The molecules with OSC less than −1 and carbon atoms more than 20 might be associated with hydrocarbon-like organic aerosol (HOA)." (page 9 lines 20-25)

*11. On page 9 lines 12-14, the authors state that the molecular weight increased during the FW events for the CHON compounds. However, all these numbers are within the reported error of each other.*

**Response:** The average molecular weights were $415 \pm 93$ Da in the NYE daytime and $472 \pm 112$ Da in the LNY daytime, which increased by about 14%. We changed the expression to be conservative: (page 10 lines 4-5)

"Their average molecular weights were $445 \pm 100$ Da in the NYE nighttime and $472 \pm 112$ Da in the LNY daytime, respectively, compared to $415 \pm 93$ Da in the NYE daytime."

*12. The paragraph on page 9 starting on line 18 is confusing. Which type of oxidized nitrogen group is being assigned for which sample? Both organonitrates and nitro- aromatics are discussed but it is unclear if these are for different samples.*

**Response:** We thank the reviewer for this comment. $N_1O_3$–$N_1O_{14}$ and $N_2O_3$–$N_2O_{13}$ subgroups were classified for CHNO compounds in all samples. The nitro-aromatics were mainly discussed among different samples in our study. To clarify, we explained the descriptions more carefully: (page 10 lines 10-12)

"CHNO compounds were classified to $N_1O_3$–$N_1O_{14}$ and $N_2O_3$–$N_2O_{13}$ subgroups in all samples

by the number of N and O atoms in their molecules (Figures 7 and S5). The total abundance of $N_1O_n$ subgroups was twice as much as that of $N_2O_n$ subgroups in each sample.

*13. The trends shown in Figure 9 are interesting and the caption is appropriately clear on how tentative these assignments are. The text that corresponds to this figure (page 9, lines 28-34) should also be adjusted to indicate that these are not structural assignments.*

**Response:** Thank you very much for this suggestion. To clarify, it has been adjusted (page 10 lines 21-23).

"…Figure 9 displays ion intensity distributions of four nitro-aromatic compounds (i.e. $C_{10}H_7O_3N$, $C_{11}H_9O_3N$, $C_{12}H_{11}O_3N$, and $C_{16}H_{79}O_3N$) detected in biomass burning aerosols by Lin et al. (2015), which were just assigned by their molecular composition but not the chemical structure. …"

*14. For the CHOS compounds, is there any reason that some of these could not be primary emissions? Have any FT-ICR studies been carried out with samples collected closer to the FW source?*

**Response:** The similar molecular composition of some of CHOS compounds derived from aliphatic, biogenic, and aromatic VOCs were detected in the FW-related aerosols, which might be evidence that they are secondary organic aerosols. We have added them in the revised manuscript as below. There have been no similar FT-ICR studies of the FW-related aerosols. Our group will try to collect these samples to answer such a good question in the future study. (on page 12 lines 22-25)

"Moreover, the aliphatic OSs of $C_{12}H_{24}O_5S$, $C_{18}H_{36}O_6S$, and $C_{10}H_{16}O_9S$, and the biogenic OSs of $C_{10}H_{18}O_5S$ and $C_{10}H_{16}O_7S$, which were separately derived from alkanes and fatty acids (Riva et al., 2016;Passananti et al., 2016;Shang et al., 2016) and $\alpha/\beta$-pinene (Surratt et al., 2008), and their corresponding family series ($C_nH_{2n}O_5S$, $C_nH_{2n}O_6S$, $C_nH_{2n-4}O_9S$, $C_{10}H_{2n-2}O_5S$ and $C_nH_{2n-4}O_7S$) were all detected in the aerosols.

…

For instance, $C_9H_{10}O_5S$, $C_{10}H_{10}O_6S$, and $C_{10}H_{10}O_7S$, derived from 2-MeNAP (Riva et al., 2015), and their corresponding family series ($C_nH_{2n-8}O_5S$, $C_nH_{2n-10}O_6S$, and $C_nH_{2n-10}O_7S$) were detected in FW-related aerosols."

*15. I recommend changing the label for the sulfur section from OSs to CHOS. This will match the rest of the paper, it will decrease confusion with OSc, and will be better given that no MS/MS studies were done to positively identify them as organosulfates (as stated on page 11).*

**Response:** As the reviewer suggested, we change some fraction of OSs to CHOS in the manuscript (on page 12 line 9, page 13 line 23). But, because organosulfates were an important fraction of CHOS species, they were tentatively assigned by the same methods as previous studies and discuss in detail. This is more conducive to its classification and the source explains.

*16. On page 12 line 12-13, the authors state: "Moreover, a great part of the FW affected ions with high intensity were potentially the BBOA". What data is this conclusion being drawn from? Is this coming from the oxidation state figure/analysis? If it is coming from the oxidation state figure, I urge caution with this type of conclusion. The carbon oxidation state is a great metric for analyzing atmospheric aging, but molecules from different sources can have similar carbon oxidation state and carbon number ranges. Please also remember that the analysis here is only looking at material that was bound and then eluted on the SPE column, is water soluble, and is easily ionized in negative ion mode. Caution should be used when making aerosol source identifications from the molecular formulas found here to ones found for different sample types with different preparation steps.*

**Response:** Thank you very much for this suggestion. The conclusion was just from the oxidation state analysis. For lack of sufficient evidence, we have removed this conclusion from the revised manuscript.

**References**

Kourtchev, I., Godoi, R. H. M., Connors, S., Levine, J. G., Archibald, A. T., Godoi, A. F. L., Paralovo, S. L., Barbosa, C. G. G., Souza, R. A. F., Manzi, A. O., Seco, R., Sjostedt, S., Park, J. H., Guenther, A., Kim, S., Smith, J., Martin, S. T., and Kalberer, M.: Molecular composition of organic aerosols in central Amazonia: an ultra-high-resolution mass spectrometry study, Atmos. Chem. Phys., 16, 11899-11913, 10.5194/acp-16-11899-2016, 2016.

Kroll, J. H., Donahue, N. M., Jimenez, J. L., Kessler, S. H., Canagaratna, M. R., Wilson, K. R., Altieri, K. E., Mazzoleni, L. R., Wozniak, A. S., and Bluhm, H.: Carbon oxidation state as a metric for describing the chemistry of atmospheric organic aerosol, Nature Chem., 3, 133-139, 2011.

Lin, P., Liu, J. M., Shilling, J. E., Kathmann, S. M., Laskin, J., and Laskin, A.: Molecular characterization of brown carbon (BrC) chromophores in secondary organic aerosol generated from photo-oxidation of toluene, Physical Chemistry Chemical Physics, 17, 23312-23325, 10.1039/c5cp02563j, 2015.

Passananti, M., Kong, L., Shang, J., Dupart, Y., Perrier, S., Chen, J., Donaldson, D. J., and George, C.: Organosulfate Formation through the Heterogeneous Reaction of Sulfur Dioxide with Unsaturated Fatty Acids and Long-Chain Alkenes, Angew. Chem. Int. Ed., 55, 10336-10339, 2016.

Riva, M., Tomaz, S., Cui, T., Lin, Y.-H., Perraudin, E., Gold, A., Stone, E. A., Villenave, E., and Surratt, J. D.: Evidence for an unrecognized secondary anthropogenic source of organosulfates and sulfonates: Gas-phase oxidation of polycyclic aromatic hydrocarbons in the presence of sulfate aerosol, Environ. Sci. Technol., 49, 6654-6664, 2015.

Riva, M., Silva Barbosa, T. D., Lin, Y.-H., Stone, E. A., Gold, A., and Surratt, J. D.: Chemical characterization of organosulfates in secondary organic aerosol derived from the photooxidation of alkanes, Atmos. Chem. Phys., 16, 11001-11018, 2016.

Shang, J., Passananti, M., Dupart, Y., Ciuraru, R., Tinel, L., Rossignol, S. p., Perrier, S. b., Zhu, T., and George, C.: SO2 Uptake on oleic acid: A new formation pathway of organosulfur compounds in the atmosphere, Environ. Sci. Technol. Let., 3, 67-72, 2016.

Surratt, J. D., Gómez-González, Y., Chan, A. W., Vermeylen, R., Shahgholi, M., Kleindienst, T. E., Edney, E. O., Offenberg, J. H., Lewandowski, M., and Jaoui, M.: Organosulfate formation in biogenic secondary organic aerosol, J. Phys. Chem. A, 112, 8345-8378, 2008.

---

## Author Comment (AC2) · 27 Apr 2020

**Authors' Responses to Reviewer #2**

We appreciate the detailed and constructive comments and suggestions from the reviewers.
The point-to-point responses to the comments are listed as below.

The *Reviewer comments are black italic font* and the Author responses are blue font.

*Xi et al. propose a study on the characterization of ambient aerosols using an FT-ICR. 6 samples were analyzed and compared to evaluate the impact of firework on air quality. Overall the data reported in this study are coherent and the structure of the paper is clear. However, additional information should be added as well as some explanation to make this paper more comprehensive.*

**Response:** We thank the reviewer for his/her comments. We have tried our best to modify and improve the quality of our manuscript.

*Page 2, lines 8-9: please reformulate.*

**Response:** The sentence has been reformulated: (page 2 lines 8-9)

"Moreover, real-time chemical composition measurements illustrated that FW-related organics were mainly secondary organic material (Jiang et al., 2015)."

*Page 2, line 31: Please provide more information regarding the sampling of the aerosol: the size of the particles; i.e., PM1, 2.5, 10? high-volume samplers?*

**Response:** Total suspended particles were collected by a high-volume sampler with pre-combusted quartz filters. To clarify, we adjust the statement in the manuscript (page 3 lines 12-14):

"Total suspended particle (TSP) sampling was conducted on the roof of a building (8 m above ground level) in the campus of the Institute of Atmospheric Physics, Chinese Academy of Sciences (39°58′28″ N, 116°22′13″ E), a representative urban site in the central north part of Beijing. TSP samples…"

*Page 3, line 2-3: I recommend the authors to use a simpler naming system. e.g., before-FW-1, after-FW-2, during-FW-1,... it would be much easier to follow the discussion.*

**Response:** Thank you very much for this suggestion. We once used your suggested method as the sample name before. Both daytime and nighttime samples were discussed in the present

study. Thus, we want to keep our expression in the current manuscript as it is.

*Page 4, line 1: The authors should explain why these samples were analyzed only in negative mode.*

**Response:** We thank the reviewer for this comment. It was because that we mainly analyzed the water-soluble fraction in aerosols, which is easily ionized in negative ion mode. The explanation has been added in the revised manuscript. (on page 4 lines 14-15)

**"Because the target species were water-soluble polar compounds, all the samples were analyzed in the negative ionization mode…"**

*Page 4, section 2.4: - why did the authors choose an S/N> 6, which is more restricting? Why not using an S/N ratio > 3, which is commonly defined as LoD?*

**Response:** The main purpose of increasing S/N ratio is to reduce the interference of noise peak and improve the reliability of molecular formula identification.

*- Is the peak assignment perform before or after blank subtraction? While the authors mentioned that blank filters were collected and analyzed, no information is providing regarding how the blank samples were used for the data analyzing.*

**Response:** We thank the reviewer for this comment. The peak assignment for real samples performed after the blank subtraction. As the reviewer suggested, the information has been added in the manuscript. (page 4 line 32-page 5 line 1)

"The molecular formulas in blank filters with a signal-to-noise ratio greater than that of the aerosol samples were subtracted from the real aerosol samples."

*- Why did the authors start at m/z 185 rather than m/z 100 as mentioned earlier in the manuscript? A significant amount of potential OA compounds can be missing.*

**Response:** For the FT-ICR MS analysis, the lower mass cut-off for the ion trap was 100 Da. To get more precise and as much information as possible about the compounds, some previous studies mainly aimed at the compounds of m/z 100 ~ 400, but relatively high molecular weight compounds of m/z 200 ~ 700 were analyzed in this study.

*- The authors should not claim any semi-quantitative results as the sensitivity of the ESI is*

*extremely dependent on the functional groups of individual compounds as reported in many studies.*

**Response:** The peak intensities of the ions was a sample to sample comparison among different samples and not a compound to compound comparison within a sample. The intensity of same compounds with similar functional groups were compared among various samples. To clarify, we have changed our state in the manuscript as reviewer #1 suggested (page 6 line 32 - page 7 line 2):

"…the peak intensities of the same ions could be compared among different samples by assuming that matrix effects were relatively constant."

*Page 5, lines 26-28: The authors should provide either some references or some sup- porting information to support their statement.*

**Response:** Thank you. Some related references have been provided with the statement in the manuscript (page 6 lines 15-18).

"Fossil fuel combustion and vehicle emissions have been reported as important sources of $NO_3^-$ in Beijing (Ianniello et al., 2010;Wang et al., 2014), while these sources minimized due to a sharp decline in the population and vehicle; most of the people leave Beijing for their hometowns during the Spring Festival (Yang et al., 2014;Zhang et al., 2017b)."

*Page 5, lines 29: How do the authors know the distribution of the ions/compounds as a function of the size provide either reference or supporting information.*

**Response:** Some related references have been provided with the statement in the manuscript (on page 6 lines 17-19).

"In addition, the concentrations of $Mg^{2+}$ and $Ca^{2+}$ were slightly higher in the NYE nighttime than the non-FW periods. They were mainly in the coarse particle mode (Huang et al., 2013;Xu et al., 2015)."

*Page 6, lines 2-4: While the concentrations of $SO_4^{2-}$, $Cl^-$ and $K^+$ are significantly different, this is not really the case for WSOC: there is an increase ∼ a factor of 2, but what's the daily variability? It is hard to conclude that FW produces a sharp increase. In other words, is the increase statistically different?*

**Response:** The original expression was overstated. We have adjusted the presentation of the

content (on page 6 lines 21-23).

"Simultaneously, the WSOC concentration peaked sharply in the NYE nighttime. Moreover, the WSOC/OC ratio was higher during the FW period than non-FW periods, indicating more water-soluble OC was formed during the FW event."

*Page 6, line 12: How do the authors know the ionization efficiency of the observed compounds? i.e., the concentrations of some ions can be very high but with very poor ionization efficiency. This statement is purely speculative*

**Response:** The statement was not very appropriate as the reviewer mentioned. The water-soluble fraction eluted from the SPE column is easily ionized in negative ion mode. To be more careful, we have adjusted the presentation of the content (on page 6 lines 30-31).

"ESI is sensitive to polar compounds, and the compounds reported in this study is easily ionized in the negative ion mode (Qi et al., 2020)."

*Page 6, lines 13-14: That's an incorrect assumption/statement. The matrix effect is one aspect. I strongly encourage the authors to check basic studies on ESI and revise their manuscript. Indeed the peak intensity can be impacted by the matrix but also depends on the ionization efficiency of individual compounds which is based on the functional groups present in each compound.*

**Response:** The peak intensity can be impacted by the ionization efficiency of individual compounds, which is based on the functional groups present in each compound. This is a sample to sample comparison for the same compounds and not that ion intensities for different compounds within a sample were compared. To clarify, we have made our statement more carefully in the manuscript (page 6 line 32 - page 7 line 2).

"…the peak intensities of the same ions could be compared among different samples by assuming that matrix effects were relatively constant (Kourtchev et al., 2016;Lin et al., 2012a)."

*Page 6, line 19: This is actually surprising and not consistent with "normal" product distribution. Indeed in most of the previous studies, most of the identified ions are between 150-250 (i.e., monomers type) and a second mode is present between 300- 400 (dimers or high molecular weight compounds), see the previous characterization using ESI-MS (QTOF, Orbitrap, and FT-ICR). The authors should comment on such a curious product distribution.*

*The authors should clearly mention that probably the vast majority of the compounds were lost during the sampling preparation.*

**Response:** Thank you very much for this suggestion. There were some reasons for the differences between some previous studies and the present manuscript. Firstly, the distribution of monomers with m/z 150-250 and dimers with m/z 300-400 mainly aimed at compounds derived from low molecular weight precursors, such as α/β-pinene. The distribution was observed in Zhang et al. (2017a) for highly oxygenated multifunctional compounds in α-Pinene secondary organic aerosol, and Romonosky et al. (2017) for aqueous photochemistry of secondary organic aerosol of α-pinene and α-humulene. But our study looked at the complex mixture of the organic compounds in ambient aerosols. There is no such a single mass spectrogram distribution for them, as the same to the previous researches (Jiang et al., 2016;Tao et al., 2014;Lin et al., 2012b). Secondly, the Ultra High Resolution mass spectrometry such as QTOF, Orbitrap, and FT-ICR MS analysis has a different mass range and center of mass due to the differences of resolution and quality accuracy. Most of the identified ions in previous researches mentioned above were between 150-500 Da, and the high intensity peaks were between 200-300 Da or there was no obvious high intensity peak range. The majority ranges of compounds in our study analyzed by 15.0 T FT-ICR MS were from 150 to 700 Da, with the high intensity peaks between 250 and 450 Da. Finally, as the reviewer mentioned, it worth noting that some fractions of compounds might be lost during the sampling preparation, particularly for the low molecular weight ones. This note has been added in the revised manuscript. (on page 7 lines 5-9)

"Thousands of formulae (~6000–9500) were obtained in each spectrum with the majority ranged from 150 to 700 Da. The molecular weights of formulae with high intensity primarily distributed between 300 and 400 Da, which were higher than previous studies with 200 - 300 Da. On the one hand, the compounds being explored in present study have a larger mass range; on the other hand, it was worth noting that some fractions of compounds might be lost during our sampling preparation, particularly for the low molecular weight ones."

*What is the reason to remove a major fraction of the organic compounds during the sampling preparation (i.e., page 3, lines 26-27)?*

**Response:** In ambient aerosols the organic components are mixed with inorganic constituents (e.g., ammonium, sulfate, nitrate, and sodium ions), which are abundant and greatly exceed the

concentrations of individual organics. Because of the special ionization mode of ESI FT-ICR MS, the presence of inorganic ions will interfere with the ion source and affect the detection results. Therefore, the inorganic salt ions should be removed from the extracts before they enter the instrumental analysis. To do that, the extracts were loaded on a solid phase extraction cartridge (Oasis HLB, Waters, USA) for desalting, which have been applied in most previous studies about ESI FT-ICR MS (Lin et al., 2012b;Bao et al., 2017;Gurganus et al., 2015;Yassine et al., 2014;Jiang et al., 2016;Mazzoleni et al., 2010). Simultaneously, some fraction of low molecular weight organic molecules, and sugars were also not retained by the cartridge. But, it had little effect on the results, as the study primarily focuses on organic compounds with relatively large molecular weights by ESI FT-ICR MS (For the FT-ICR MS analysis, lower mass cut-off for the ion trap was 100 Da and the mass limit was from 185 Da to 1000 Da). The reason has been briefly added in the revised manuscript. (on page 4 lines 8-9)

"The extract was combined and loaded onto a SPE cartridge (Oasis HLB, Waters, U.S.) for desalting…".

*Page 6, line 33 and page 7, lines 1-2: This is overstated, the authors should provide deeper statistical analysis before making such a statement. Are the numbers really statistically different?*

**Response:** Thank you. The statements seem a bit decisive. To be more carefully, the conclusion has been made more rigorous. (on page 7 lines 23-25)

"These results suggested that FW emission contributes the formation of relatively high molecular weight compounds in urban aerosols. In addition, the average DBE values, an indicative of degree of unsaturation, increased from $9.35 \pm 4.01$ in the NYE daytime to $10.1 \pm 4.82$ in the NYE nighttime and $11.2 \pm 4.98$ in the LNY daytime."

*Page 7, line 8: Please provide numbers to support such a statement.*

**Response:** Thank you very much for this suggestion. The detailed numbers of high molecular weight compounds have been provided. (on page 7 lines 32-33, page 8 lines 1-2)

"…, FW emission dramatically increased the amounts of HMW (>400 Da) organic compounds from 3022 compounds in the NYE daytime to 4264 compounds in the NYE nighttime and 5206 compounds in the LNY daytime, while the relative abundance of three categories compounds were different."

*Page 7, lines 12-14: I am confused by this sentence. Why did the authors refer to Ms2 studies while they didn't perform such an analysis.*

**Response:** We thank the reviewer for this comment. We agree that this sentence is redundant here and may cause some confusion. It was deleted in the revised manuscript. The original sentence was only to explain that there were different types of CHO compounds according previous studies. And the main content of this article was not to discuss some individual compound structures. So, MS/MS studies have not included in the study.

*Page 7, line 30: This statement is incorrect. It is not in urban aerosol but in the filter extract, i.e., after removing a major fraction of organic aerosol components.*

**Response:** We thank the reviewer for this comment. The filter extract was the fraction of water-soluble organic matter in urban aerosol. To clarify, the statement has been adjusted more carefully (on page 8 lines 19-21):

"As shown in Figure 4, the high intensity CHO compounds in the fraction of water-soluble organic matter in urban aerosols are primarily with C numbers of 15–27 and DBE values of 6–15, indicating that they potentially have one or more benzene rings in their molecules."

*Page 9, lines 20-27: This section is confusing. please clarify, i.e., within the same paragraph the authors claim that CHON with O>3 are likely organonitrates and a few lines below nitroaromatics. In addition, the authors should keep in mind the difference in terms of ionization efficiency of such compounds: i.e., compounds containing nitro-functional groups have a very high ionization efficiency (e.g., nitroaromatics), unlike organonitrate compounds.*

**Response:** We thank the reviewer's comments. The nitro-aromatics were mainly discussed among different samples in our study. To clarify, the confusing parts were removed in the revised manuscript.

*Why is the CHONS group not discussed in the paper?*

**Response:** CHONS compounds are obviously subject to nighttime chemical oxidation. Due to the limited length of the paper, their characteristics were discussed in detail in another manuscript, especially for the investigation of the volatility of CHONS estimated by a molecular corridor method. To clarify, we have added the following sentence to the manuscript

in the end of the introduction as Reviewer #1 suggested (on page 3 lines 8-9):

"In addition, the detailed molecular characteristics of CHNOS species and their volatility using a molecular corridor method will be present in another study."

**References:**

Bao, H. Y., Niggemann, J., Luo, L., Dittmar, T., and Kao, S. J.: Aerosols as a source of dissolved black carbon to the ocean, Nat. Commun., 8, 7, 10.1038/s41467-017-00437-3, 2017.

Gurganus, S. C., Wozniak, A. S., and Hatcher, P. G.: Molecular characteristics of the water soluble organic matter in size-fractionated aerosols collected over the North Atlantic Ocean, Mar. Chem., 170, 37-48, 2015.

Huang, Y., Liu, Z., Chen, H., and Wang, Y.: Characteristics of mass size distributions of water-soluble, inorganic ions during summer and winter haze days of Beijing, Huan jing ke xue= Huanjing kexue, 34, 1236-1244, 2013.

Ianniello, A., Spataro, F., Esposito, G., Allegrini, I., Rantica, E., Ancora, M., Hu, M., and Zhu, T.: Occurrence of gas phase ammonia in the area of Beijing (China), Atmos. Chem. Phys., 10, 9487, 2010.

Jiang, B., Kuang, B. Y., Liang, Y., Zhang, J., Huang, X. H., Xu, C., Yu, J. Z., and Shi, Q.: Molecular composition of urban organic aerosols on clear and hazy days in Beijing: a comparative study using FT-ICR MS, Environ. Chem., 13, 888-901, 2016.

Jiang, Q., Sun, Y., Wang, Z., and Yin, Y.: Aerosol composition and sources during the Chinese Spring Festival: fireworks, secondary aerosol, and holiday effects, Atmos. Chem. Phys., 15, 6023-6034, 2015.

Kourtchev, I., Godoi, R. H. M., Connors, S., Levine, J. G., Archibald, A. T., Godoi, A. F. L., Paralovo, S. L., Barbosa, C. G. G., Souza, R. A. F., Manzi, A. O., Seco, R., Sjostedt, S., Park, J. H., Guenther, A., Kim, S., Smith, J., Martin, S. T., and Kalberer, M.: Molecular composition of organic aerosols in central Amazonia: an ultra-high-resolution mass spectrometry study, Atmos. Chem. Phys., 16, 11899-11913, 10.5194/acp-16-11899-2016, 2016.

Lin, P., Rincon, A. G., Kalberer, M., and Yu, J. Z.: Elemental composition of HULIS in the Pearl River Delta Region, China: Results inferred from positive and negative electrospray high resolution mass spectrometric data, Environ. Sci. Technol., 46, 7454-7462, 2012a.

Lin, P., Yu, J. Z., Engling, G., and Kalberer, M.: Organosulfates in humic-like substance fraction isolated from aerosols at seven locations in East Asia: A study by ultra-high-resolution mass

spectrometry, Environ. Sci. Technol., 46, 13118-13127, 2012b.

Mazzoleni, L. R., Ehrmann, B. M., Shen, X., Marshall, A. G., and Collett Jr, J. L.: Water-soluble atmospheric organic matter in fog: exact masses and chemical formula identification by ultrahigh-resolution Fourier transform ion cyclotron resonance mass spectrometry, Environ. Sci. Technol., 44, 3690-3697, 2010.

Qi, Y. L., Fu, P. Q., Li, S. L., Ma, C., Liu, C. Q., and Volmer, D. A.: Assessment of molecular diversity of lignin products by various ionization techniques and high-resolution mass spectrometry, Science of the Total Environment, 713, 10.1016/j.scitotenv.2020.136573, 2020.

Romonosky, D. E., Li, Y., Shiraiwa, M., Laskin, A., Laskin, J., and Nizkorodov, S. A.: Aqueous Photochemistry of Secondary Organic Aerosol of α-Pinene and α-Humulene Oxidized with Ozone, Hydroxyl Radical, and Nitrate Radical, J. Phys. Chem. A, 121, 1298-1309, 2017.

Tao, S., Lu, X., Levac, N., Bateman, A. P., Nguyen, T. B., Bones, D. L., Nizkorodov, S. A., Laskin, J., Laskin, A., and Yang, X.: Molecular characterization of organosulfates in organic aerosols from Shanghai and Los Angeles urban areas by nanospray-desorption electrospray ionization high-resolution mass spectrometry, Environ. Sci. Technol., 48, 10993-11001, 2014.

Wang, Y., Yao, L., Wang, L., Liu, Z., Ji, D., Tang, G., Zhang, J., Sun, Y., Hu, B., and Xin, J.: Mechanism for the formation of the January 2013 heavy haze pollution episode over central and eastern China, Science China Earth Sciences, 57, 14-25, 2014.

Xu, J., Zhang, Q., Wang, Z., Yu, G., Ge, X., and Qin, X.: Chemical composition and size distribution of summertime PM2. 5 at a high altitude remote location in the northeast of the Qinghai–Xizang (Tibet) Plateau: insights into aerosol sources and processing in free troposphere, Atmos. Chem. Phys, 15, 5069-5081, 2015.

Yang, L., Gao, X., Wang, X., Nie, W., Wang, J., Gao, R., Xu, P., Shou, Y., Zhang, Q., and Wang, W.: Impacts of firecracker burning on aerosol chemical characteristics and human health risk levels during the Chinese New Year Celebration in Jinan, China, Sci. Total Environ., 476, 57-64, 2014.

Yassine, M. M., Harir, M., Dabek-Zlotorzynska, E., and Schmitt-Kopplin, P.: Structural characterization of organic aerosol using Fourier transform ion cyclotron resonance mass spectrometry: aromaticity equivalent approach, Rapid Commun. Mass Spectrom., 28, 2445-2454, 2014.

Zhang, X., Lambe, A. T., Upshur, M. A., Brooks, W. A., Be, A. G., Thomson, R. J., Geiger, F. M., Surratt, J. D., Zhang, Z. F., Gold, A., Graf, S., Cubison, M. J., Groessl, M., Jayne, J. T.,

Worsnop, D. R., and Canagaratna, M. R.: Highly Oxygenated Multifunctional Compounds in alpha-Pinene Secondary Organic Aerosol, Environmental Science & Technology, 51, 5932-5940, 10.1021/acs.est.6b06588, 2017a.

Zhang, Y., Wei, J., Tang, A., Zheng, A., Shao, Z., and Liu, X.: Chemical characteristics of PM2.5 during 2015 spring festival in Beijing, China, Aerosol and Air Quality Research, 17, 1169-1180, 2017b.

---

## Author Comment (AC3) · 27 Apr 2020

**Authors' Responses to Reviewer #3**

We appreciate the detailed and constructive comments and suggestions from the reviewers. The point-to-point responses to the comments are listed as below.

The *Reviewer comments are black italic font* and the Author responses are blue font.

*General Comments:*

*The article studied the characteristic of CHO, CHNO, and CHOS before, during, and after FW event. Many species were detected by FT-ICR MS, and were analyzed through the manuscript. Furthermore, potential sources of these subgroups were also discussed through many calculations. There are many data and analysis methods which help the reader to understand the different pollution characters of aerosols during six periods. The whole article was aimed to discuss the event of firework-related urban aerosols before, during, and after New Year's Eve evening. The author just discussed the redox chemistry driven by NOx, O3, and OH, but the impact of combustion processes during the FW event wasn't discussed. What the relation between combustion processes and the three subgroups? I think the combustion processes is an important factor for the pollution during FW event. For example, there are amount of sulfur in the firework which many contribute the formation of CHOS species. Besides, the meteorology parameters were not contained in the article, which is hardly to analysis the sources of these subgroups studied in the article. for example, the author indicated the "multiphase" redox chemistry is important for the detected species formation, but how about the RH during these periods? The article should be revised according the comments and then can be published.*

**Response:** We really appreciate the valuable comments from the reviewer. The impact of combustion processes during the FW event was discussed as the reviewer comments below (on page 10 lines 24-28; page 11 lines 8-15). Unfortunately, meteorology data were not collected during that period. Thank you very much for giving us such a good suggestion. We will pay attention to collecting meteorological data in the future to discuss the content more comprehensively.

"Nitro-aromatic compounds are produced in the atmosphere via the oxidation of aromatic precursors in the presence of $NO_2$ (Laskin et al., 2015), and their relative yields increase with $NO_2/NO_3$ concentrations (Sato et al., 2007;Jang and Kamens, 2001), which can be released in large quantities during FW combustion processes. Moreover, high abundant PAHs from FW

emission in the NYE can react more efficiently with $NO_2$ than their single-ring aromatic counterparts (Nishino et al., 2009).**"**

**"**The amount of sulfur in the firework was released into the air with the form of sulfur oxides during the combustion process, and further produced acidified sulfate seed aerosol, which considerably contributed to the formation of a large number of CHOS compounds via acid catalyzed reaction with biogenic and anthropogenic volatile organic compounds (VOCs) (Surratt et al., 2008;Riva et al., 2015). For instance, the CHOS compounds derived from monoterpenes and sesquiterpenes, such as limonene, α/γ-terpinene and β-caryophyllene, were detected only under acidic or strongly acidic sulfate seed aerosol conditions (Surratt et al., 2008;Iinuma et al., 2007a;Iinuma et al., 2007b;Chan et al., 2011). Meanwhile, the high levels of nitrogen oxides emitted by FW burning can promote the formation of some CHOS compounds (Surratt et al., 2008).**"**

*Specific Comments:*

*Introduction: Why the author studied subgroups of CHO, CHNO, and CHOS during these periods?*

**Response:** Thank you very much for this suggestion. The explanation of the importance in studied subgroups of CHO, CHNO, and CHOS have been presented in the Introduction section. (on page 2 lines 12-26)

"Water-soluble organic carbon (WSOC) is a ubiquitous component of atmospheric aerosols. A large proportion of water-soluble organic matter is composed of HMW organic compounds that contain a substantial fraction of heteroatoms (N, S, O) (Lin et al., 2012;Mazzoleni et al., 2012;Wozniak et al., 2008;Wang et al., 2016). Highly oxygenated molecules contain a wide range of chemical functional groups such as peroxides, hydroperoxides, carbonyls, and per-carboxylic acids (Lee et al., 2019). Organic acids in oxygen-containing species contribute significantly to aerosol acidity. Lots of nitro-aromatic compounds in relatively high molecular weight compounds, often observed in biomass burning aerosols, are potential contributors to light absorption (Laskin et al., 2015;Lin et al., 2015). Moreover, organosulfates substantially contribute to the secondary organic aerosol (SOA) mass (Tolocka and Turpin, 2012), which plays an important role in exploring the formation pathway of SOA (Shang et al., 2016;Riva et al., 2015;Riva et al., 2016;Passananti et al., 2016). Meanwhile, because of their polar and hydrophilic nature, organosulfates can influence the hygroscopic properties of aerosols

(Estillore et al., 2016). Hence, to characterize both the compound class and individual compound level of organic aerosols (OA) is important for exploring the formation mechanisms, physicochemical properties, and environmental effects of firework-related aerosols. Moreover, the large amount of firework emission is an ideal event to understand the contribution of anthropogenic precursors to the formation of organic aerosols."

*Page 5 Line 28-30: The author ascribed the increase of $Mg^{2+}$ and $Ca^{2+}$ to the dust particles increased were not exactly. Mg would exist in the FW. Did the author get the PM10 data?*

**Response:** We thank the reviewer for the comments. The description about $Mg^{2+}$ has been changed (on page 6 lines 17-19). PM10 were not collected in the study, but the total suspended particles (TSP) were collected and analyzed including aerosols with particle size less than 10 microns.

"In addition, the concentrations of $Mg^{2+}$ and $Ca^{2+}$ were slightly higher in the NYE nighttime than the non-FW periods. They were mainly in the coarse particle mode (Huang et al., 2013;Xu et al., 2015)."

*Page 6 Line 20: CHNOS was mentioned here and also in Figure 1. Why the author didn't discuss CHNOS? What the relationship between CHNOS with CHNO and CHOS?*

**Response:** CHONS compounds are obviously subject to nighttime chemical oxidation. Due to the limited length of the paper, their characteristics were discussed in detail with the corresponding nighttime chemistry in detail in another manuscript. To clarify, as suggested by the other reviewers, we have added the following sentence to the manuscript in the end of the introduction (page 3 lines 8-9):

"In addition, the detailed molecular characteristics of CHNOS species and their volatility using a molecular corridor method will be present in another study."

*Page 7 Line 8: while the relative abundance of "four" categories compounds: "four" or "three"?*

**Response:** Sorry for the misrepresentation. "four" has been changed to "three" in the revised manuscript. (on page 7 lines 32-33)

"……, while the relative abundance of compounds in three categories was different."

*Page 8 Line 18: Xc can help to more precisely identify and characterize aromatic and condensed aromatic compounds in highly complex WSOC mixtures, why AI method was used in the manuscript?*

**Response:** Previous studies have applied the AI method to characterize aromatic compounds for highly complex WSOC mixtures in atmospheric aerosols. For consistency, and for comparison with them, the AI method was used in the manuscript. Meanwhile, Xc was also applied to precisely identify the aromatic species in the manuscript.

*Figure 6: I can't understand this picture, the markers can't be seen clearly, the green areas can't be understood.*

**Response:** We thank the reviewer for this comment. To make it clearer, we have modified the diagram as shown below. Due to the large amount of data, we presented each sample separately in one chart. The explanatory graph was added with the lower left corner of the graph (a). The gray areas were the similar molecular composition as characterized by Kroll et al. (2011) and Kourtchev et al. (2016).

[Figure]

**Figure 6.** Overlaid carbon oxidation state (OS$_C$) symbols for CHO compounds in NYE D (a), NYE N (b), and LNY D (c) samples. The size and color bar of the markers reflects the relative peak intensities of compounds on a logarithmic scale. The gray areas were marked as SV-OOA (semi-volatile oxidized organic aerosol), LV-OOA (low-volatility oxidized organic aerosol), BBOA (biomass burning organic aerosol) and HOA (hydrocarbon-like organic aerosol) (Kourtchev et al., 2016;Kroll et al., 2011).

*Page 10 Line21-24, Page 11 Line 25: The author highlights the importance of nighttime chemical oxidation to the formation of CHOS compounds, what was the evidence? How the combustion process impacted the formation of CHOS during FW event?*

**Response:** Different from the daytime samples, the CHOS compounds considerably increased

in NYE nighttime sample with large FW emission. It indicated that there were unknown formation pathways of CHOS compounds with nighttime chemical oxidation. The combustion process of FW has been included in discussion in the manuscript. (page 11 lines 8-15)

[revised manuscript text omitted]